# Fast Projected Newton-like Method for Precision Matrix Estimation under Total Positivity

**Jian-Feng Cai**[1,2], **José Vinícius de M. Cardoso**[1], **Daniel P. Palomar**[1], **Jiaxi Ying**[1,2]*

Hong Kong University of Science and Technology[1]

HKUST Shenzhen-Hong Kong Collaborative Innovation Research Institute[2]

## Abstract

We study the problem of estimating precision matrices in Gaussian distributions that are multivariate totally positive of order two ($\mathrm{MTP}_2$). The precision matrix in such a distribution is an *M*-matrix. This problem can be formulated as a sign-constrained log-determinant program. Current algorithms are designed using the block coordinate descent method or the proximal point algorithm, which becomes computationally challenging in high-dimensional cases due to the requirement to solve numerous nonnegative quadratic programs or large-scale linear systems. To address this issue, we propose a novel algorithm based on the two-metric projection method, incorporating a carefully designed search direction and variable partitioning scheme. Our algorithm substantially reduces computational complexity, and its theoretical convergence is established. Experimental results on synthetic and real-world datasets demonstrate that our proposed algorithm provides a significant improvement in computational efficiency compared to the state-of-the-art methods.

## 1 Introduction

We consider the problem of estimating the precision matrix (*i.e.,* inverse covariance matrix) in a multivariate Gaussian distribution, where all the off-diagonal elements of the precision matrix are nonpositive. The resulting precision matrix is a symmetric *M-matrix*. Such property is also known as multivariate totally positive of order two ($\mathrm{MTP}_2$) [1]. For ease of presentation, we call the nonpositivity constraints on the off-diagonal elements of the precision matrix as $\mathrm{MTP}_2$ constraints. This model arises in a variety of applications such as taxonomic reasoning [2], graph signal processing [3], factor analysis in psychometrics [4], and financial markets [5].

Estimating precision matrices under $\mathrm{MTP}_2$ constraints is an active research topic. Recent results in [2, 4, 6] show that $\mathrm{MTP}_2$ constraints lead to a drastic reduction on the number of observations required for the maximum likelihood estimator (MLE) to exist in Gaussian distributions and Ising models. This advantage is crucial in high-dimensional regimes with limited observations. Growing interest in estimating precision matrices under $\mathrm{MTP}_2$ constraints is seen in graph signal processing [7–9]. A precision matrix fulfilling $\mathrm{MTP}_2$ constraints can be regarded as a generalized graph Laplacian, where eigenvalues and eigenvectors are interpreted as spectral frequencies and Fourier bases, facilitating the computation of graph Fourier transform [10]. The $\mathrm{MTP}_2$ property has also been studied in portfolio allocation [5] and structure recovery [11, 12].

Estimating precision matrices under $\mathrm{MTP}_2$ constraints can be formulated as a sign-constrained log-determinant program. Existing algorithms for solving this problem, such as the block coordinate descent (BCD) [2, 3, 8] and proximal point algorithm (PPA) [9], are efficient for low-dimensional

---

*Corresponding author.

problems. However, they become time-consuming in high-dimensional scenarios due to the necessity of solving numerous nonnegative quadratic programs or large-scale linear systems. An alternative is the gradient projection method [13], which offers computational efficiency per iteration. Nevertheless, it often grapples with slow convergence rates in high-dimensional cases. Thus, there is a demand for efficient and scalable algorithms for precision matrix estimation under $\mathrm{MTP}_2$ constraints.

## 1.1 Contributions

In this paper, we propose a fast projected Newton-like algorithm for estimating precision matrices under $\mathrm{MTP}_2$ constraints. While second-order algorithms generally require fewer iterations than first-order methods, they often encounter computational challenges due to the necessity of computing a large (approximate) Hessian matrix inverse or equivalently solving the linear system. Our main contributions in this paper are threefold:

1. Our proposed algorithm, rooted in the two-metric projection method, stands apart from current BCD and PPA approaches [2, 3, 8, 9] for solving the target problem. Utilizing a well-designed search direction and variable partitioning scheme, our algorithm avoids the need to solve nonnegative quadratic programs or linear systems, yielding a significant computational reduction compared to BCD and PPA algorithms. As a second-order method, our algorithm maintains the same per-iteration computational complexity as the gradient projection method.

2. We establish that our algorithm converges to the minimizer of the target problem. Furthermore, under a mild assumption, we prove the convergence of the set of *free* variables to the support of the minimizer within finite iterations and provide the convergence rate of our algorithm.

3. Numerical experiments on both synthetic and real-world datasets provide compelling evidence that our algorithm converges to the minimizer considerably faster than the compared methods. We apply the proposed method to financial time-series data and observe significant performance in terms of *modularity* value on the learned financial networks.

**Notation:** Lower case bold letters denote vectors and upper case bold letters denote matrices. Both $X_{ij}$ and $[\boldsymbol{X}]_{ij}$ denote the $(i,j)$-th entry of $\boldsymbol{X}$. $[p]$ denotes the set $\{1, \ldots, p\}$, and $[p]^2$ denotes the set $\{1, \ldots, p\} \times \{1, \ldots, p\}$. Let $\otimes$ be the Kronecker product and $\odot$ be the entry-wise product. $\mathrm{supp}(\boldsymbol{X}) = \{(i,j) \in [p]^2 \,|\, X_{ij} \neq 0\}$. $\|\boldsymbol{X}\|_{\max} = \max_{i,j} |X_{ij}|$ and $\|\boldsymbol{X}\|_{\min} = \min_{i,j} |X_{ij}|$. $\mathbb{S}_+^p$ and $\mathbb{S}_{++}^p$ denote the sets of symmetric positive semi-definite and positive definite matrices with dimension $p \times p$. $\mathrm{vec}(\boldsymbol{X})$ and $\boldsymbol{X}^\top$ denote the vectorized version and transpose of $\boldsymbol{X}$.

## 2 Problem Formulation and Related Work

In this section, we first introduce the problem formulation, then present related works.

### 2.1 Problem formulation

Let $\boldsymbol{y} = (y_1, \ldots, y_p)$ be a $p$-dimensional random vector following $\boldsymbol{y} \sim \mathcal{N}(\boldsymbol{0}, \boldsymbol{\Sigma})$, where $\boldsymbol{\Sigma}$ is the covariance matrix. We focus on the problem of estimating the precision matrix $\boldsymbol{\Theta} := \boldsymbol{\Sigma}^{-1}$ given $n$ i.i.d. observations $\boldsymbol{y}^{(1)}, \ldots, \boldsymbol{y}^{(n)}$. Let $\boldsymbol{S} = \frac{1}{n} \sum_{i=1}^n \boldsymbol{y}^{(i)} \big(\boldsymbol{y}^{(i)}\big)^\top$ be the sample covariance matrix. Throughout the paper, the sample covariance matrix is assumed to have strictly positive diagonal elements, which holds with probability one. This is because some diagonal element $S_{jj}$ is zero if and only if the $j$-th element of $\boldsymbol{y}^{(i)}$ must be zero for every $i \in [p]$, which holds with probability zero.

We consider solving the following sign-constrained log-determinant program:

$$
\begin{aligned}
\boldsymbol{X}^\star := \quad &\underset{\boldsymbol{X} \in \mathcal{M}^p}{\arg\min} \quad -\log\det(\boldsymbol{X}) + \mathrm{tr}(\boldsymbol{X}\boldsymbol{S}) + \textstyle\sum_{i \neq j} \lambda_{ij} |X_{ij}|, \\
&\text{subject to} \quad X_{ij} = 0, \ \forall \, (i,j) \in \mathcal{E},
\end{aligned}
\tag{1}
$$

where $\lambda_{ij}$ is the regularization parameter, $\mathcal{E}$ is the disconnectivity set with each node pair forced to disconnect, and $\mathcal{M}^p$ is the set of all $p$-dimensional, symmetric, non-singular *M-matrices* defined by

$$
\mathcal{M}^p := \big\{ \boldsymbol{X} \in \mathbb{S}_{++}^p \,|\, X_{ij} \leq 0, \ \forall \, i \neq j \big\}.
\tag{2}
$$

The disconnectivity set in (1) can be obtained in several ways: (i) it is often the case that some edges between nodes must not exist due to prior knowledge; (ii) it can be estimated from initial estimators; (iii) it can be obtained in some tasks of learning structured graphs such as bipartite graph [14–16].

## 2.2 Related work

Estimating precision matrices under Gaussian graphical models has been extensively studied in the literature. One well-known method is graphical lasso [17–20], which minimizes the $\ell_1$-regularized Gaussian negative log-likelihood. Various algorithms were proposed to solve this problem including first-order methods [17, 18, 21–29] and second-order methods [30–33]. The graphical lasso optimization problem is unconstrained and nonsmooth, while Problem (1) is smooth and constrained. The difficulties in solving the two problems are inherently different, and the algorithms mentioned above cannot be directly extended to solve Problem (1).

Recent studies [2, 3, 8, 9] employed BCD and PPA-type algorithms to estimate precision matrices under $\mathrm{MTP}_2$ constraints. In [2], the primal variable is updated one column/row at a time by solving a nonnegative quadratic program, cycling until convergence. The work [8] follows a similar approach but addresses the dual problem, improving memory efficiency. Both works target Problem (1) without the disconnectivity constraint. A BCD-type algorithm was proposed in [3] to accommodate disconnectivity constraints. However, the computational complexity of these algorithms, at $O(p^4)$ operations per cycle, becomes prohibitive for high-dimensional problems. Alternatively, recent work [9] introduced an inexact PPA algorithm to solve a transformed problem, derived using the soft-thresholding technique. However, this algorithm demands the computation of an inexact Newton direction from a $p^2 \times p^2$ linear system at every iteration within the inner loop, presenting computational difficulties in high-dimensional scenarios.

The proposed algorithm adopts the two-metric projection framework [34], incorporating distinct metrics for search direction and projection. A representative method in this framework is the projected Newton algorithm [35], originally designed for nonnegativity constrained problems. However, it is unsuitable for Problem (1) due to its $O(p^6)$ operations needed to compute the inverse of the Hessian at each iteration. To mitigate computation and memory costs, the projected quasi-Newton algorithm with limited-memory Boyden-Fletcher-Goldfarb-Shanno (L-BFGS) was introduced in [36], requiring $O((m + p)p^2)$ operations per iteration, with $m$ the number of iterations stored for Hessian approximation. By leveraging the structure of Problem (1), this paper carefully designs the search direction and variable partitioning scheme to substantially reduce computation and memory costs, achieving the same orders per iteration as the gradient projection method [13].

## 3 Proposed Algorithm

In this section, we propose a fast projected Newton-like algorithm to solve Problem (1). The constraints in Problem (1) can be rewritten as $\boldsymbol{X} \in \Omega \cap \mathbb{S}_{++}^p$, where $\Omega$ is defined as

$$\Omega := \left\{ \boldsymbol{X} \in \mathbb{R}^{p \times p} \,|\, X_{ij} = 0, \, \forall \,(i, j) \in \mathcal{E}; \; X_{ij} \leq 0, \, \forall \, i \neq j \right\}.$$

The set $\Omega$ is convex and closed, thus this constraint can be handled by a projection $\mathcal{P}_\Omega$ defined by

$$\left[ \mathcal{P}_\Omega(\boldsymbol{A}) \right]_{ij} = \begin{cases} 0 & \text{if } (i, j) \in \mathcal{E}, \\ A_{ij} & \text{if } i = j, \\ \min(A_{ij}, 0) & \text{if } (i, j) \notin \mathcal{E} \text{ and } i \neq j. \end{cases} \tag{3}$$

The positive definite set $\mathbb{S}_{++}^p$ is not closed and cannot be managed by a projection, which will be handled using the line search method in Section 3.3. Let $f$ denote the objective function of Problem (1). To address Problem (1), we start with the gradient projection method, expressed as:

$$\boldsymbol{X}_{k+1} = \mathcal{P}_\Omega \big( \boldsymbol{X}_k - \gamma_k \nabla f\left( \boldsymbol{X}_k \right) \big), \tag{4}$$

where $\gamma_k$ is the step size. To accelerate convergence, one may consider

$$\boldsymbol{X}_{k+1} = \mathcal{P}_\Omega \big( \boldsymbol{X}_k - \gamma_k \boldsymbol{P}_k \big), \tag{5}$$

in which $\boldsymbol{P}_k \in \mathbb{R}^{p \times p}$ is a search direction defined by

$$\mathrm{vec}\left( \boldsymbol{P}_k \right) = \boldsymbol{M}_k^{-1} \mathrm{vec}\left( \nabla f(\boldsymbol{X}_k) \right), \tag{6}$$

where $\boldsymbol{M}_k \in \mathbb{R}^{p^2 \times p^2}$ is a positive definite symmetric matrix, incorporating second-order derivative information. If we adopt $\boldsymbol{M}_k$ as the Hessian matrix, then $\boldsymbol{P}_k$ becomes the Newton direction as shown in Proposition 3.1, with the proof provided in Appendix B.1, and iterate (5) can be viewed as a natural adaptation of the unconstrained Newton's method. Regrettably, the convergence of such an iterate to the minimizer cannot be guaranteed, as $\boldsymbol{P}_k$ may not be a descent direction here, which is supported by numerical results in Section 5.1.1. Similar observations have also been reported in [35, 37].

**Proposition 3.1.** *If $M_k$ is constructed as the Hessian matrix $H_k$ of Problem* (1)*, then the search direction $P_k$ defined in* (6) *can be written as $P_k = X_k \nabla f(X_k) X_k$.*

*Remark* 3.2. We refer to iterate (5) as the two-metric projection method [34], as it adopts two distinct metrics: the search direction $P_k$ induced by the quadratic norm $\|\cdot\|_{M_k}$ defined by $\|A\|_{M_k} = \langle \operatorname{vec}(A),\ M_k \operatorname{vec}(A) \rangle^{\frac{1}{2}}$, and the projection $\mathcal{P}_\Omega$ with respect to the Frobenius norm $\|\cdot\|_{\mathrm{F}}$.

## 3.1  Identifying the sets of *restricted* and *free* variables

In order to guarantee iterate (5) to converge to the minimizer, we partition the variables into two groups, *i.e.*, *restricted* and *free* variables, and update the two groups separately. We first define a set $\mathcal{T}(X, \epsilon)$ with respect to $X \in \mathbb{S}^p_{++}$ and $\epsilon \in \mathbb{R}_+$,

$$\mathcal{T}(X, \epsilon) := \big\{ (i,j) \in [p]^2 \,|\, -\epsilon \le X_{ij} \le 0,\ [\nabla f(X)]_{ij} < 0 \big\}. \tag{7}$$

Then at the $k$-th iteration, we identify the set of *restricted* variables based on $X_k$ as follows,

$$\mathcal{I}_k := \mathcal{T}(X_k, \epsilon_k) \cup \mathcal{E}, \tag{8}$$

where $\mathcal{E}$ is the disconnectivity set from Problem (1), and $\epsilon_k$ is a small positive scalar. For any $(i,j) \in \mathcal{T}(X_k, \epsilon_k)$, $X_{ij}$ in the next iterate is likely to be outside the feasible set (*i.e.*, $X_{ij} > 0$) if we remove the projection $\mathcal{P}_\Omega$, as it is near zero and moves towards the positive direction if using the negative of the gradient as the search direction. Therefore, we set all *restricted* variables to zero.

To establish the theoretical convergence of the algorithm, the positive scalar $\epsilon_k$ in (8) is specified as

$$\epsilon_k := \min \Big( 2(1-\alpha)m^2 \big\| [\nabla f(X_k)]_{\mathcal{T}_\delta \setminus \mathcal{E}} \big\|_{\min},\ \delta \Big), \tag{9}$$

where $m$ is a positive scalar (See Lemma B.1 in Appendix), $\alpha \in (0,1)$ is a parameter in the line search condition, and $\mathcal{T}_\delta$ represents the set $\mathcal{T}(X_k, \delta)$. In the rare event that $\mathcal{T}_\delta$ is empty, particularly in sparse settings, we define $\epsilon_k = \delta$, implying an empty $\mathcal{T}(X_k, \epsilon_k)$ in (8). The parameter $\delta$ satisfies $0 < \delta < \min_{(i,j) \in \operatorname{supp}(X^\star)} |[X^\star]_{ij}|$, where $X^\star$ is the minimizer of Problem (1). Such a condition can be ensured by setting a sufficiently small positive $\delta$. Then $\epsilon_k$ in (9) is nearly equal to $\delta$. From an implementation view, we can directly set a small positive $\epsilon_k$, resulting in the algorithm performing well in practice. The set of free variables, denoted by $\mathcal{I}_k^c$, is the complement of $\mathcal{I}_k$.

## 3.2  Computing approximate Newton direction

While the (approximate) Newton direction usually demands a considerably higher computational cost than the gradient, our designed direction maintains the same computational order as the gradient.

At the $k$-th iteration, we first partition $X_k$ into two groups, $[X_k]_{\mathcal{I}_k}$ and $[X_k]_{\mathcal{I}_k^c}$, where $[X_k]_{\mathcal{I}_k} \in \mathbb{R}^{|\mathcal{I}_k|}$ and $[X_k]_{\mathcal{I}_k^c} \in \mathbb{R}^{|\mathcal{I}_k^c|}$ denote two vectors containing all elements of $X_k$ in the sets $\mathcal{I}_k$ and $\mathcal{I}_k^c$, respectively. Then we can rewrite the search direction $P_k$ in (6) as follows,

$$\operatorname{pvec}_k(P_k) = Q_k \operatorname{pvec}_k(\nabla f(X_k)), \tag{10}$$

where $\operatorname{pvec}_k(P_k)$ stacks $P_k$ into a vector, similar to $\operatorname{vec}(P_k)$, but places elements from $\mathcal{I}_k^c$ first, followed by those from $\mathcal{I}_k$. $Q_k$ is obtained by permuting the rows and columns of $M_k^{-1}$ in (6). To enhance computational efficiency, we propose constructing $Q_k$ and $M_k^{-1}$ as follows:

$$Q_k = \begin{bmatrix} [M_k^{-1}]_{\mathcal{I}_k^c \mathcal{I}_k^c} & [M_k^{-1}]_{\mathcal{I}_k^c \mathcal{I}_k} \\ [M_k^{-1}]_{\mathcal{I}_k \mathcal{I}_k^c} & [M_k^{-1}]_{\mathcal{I}_k \mathcal{I}_k} \end{bmatrix} = \begin{bmatrix} [H_k^{-1}]_{\mathcal{I}_k^c \mathcal{I}_k^c} & 0 \\ 0 & D_k \end{bmatrix}, \tag{11}$$

where $D_k \in \mathbb{R}^{|\mathcal{I}_k| \times |\mathcal{I}_k|}$ is a positive definite diagonal matrix, and $[H_k^{-1}]_{\mathcal{I}_k^c \mathcal{I}_k^c} \in \mathbb{R}^{|\mathcal{I}_k^c| \times |\mathcal{I}_k^c|}$ is a principal submatrix of $H_k^{-1}$, preserving rows and columns indexed by $\mathcal{I}_k^c$. Here, $H_k$ is the Hessian matrix at $X_k$. The construction of $M_k^{-1}$ in (11) is crucial for defining the search direction, enabling computation and memory costs comparable to the gradient projection method while effectively incorporating second-order derivative information.

Next, we compute the approximate Newton direction $P_k$ over the set $\mathcal{I}_k^c$ and present the iterate $[X_{k+1}]_{\mathcal{I}_k^c}$. We define a projection $\mathcal{P}_{\mathcal{I}_k^c}(A)$ as follows,

$$\big[ \mathcal{P}_{\mathcal{I}_k^c}(A) \big]_{ij} = \begin{cases} A_{ij} & \text{if } (i,j) \in \mathcal{I}_k^c, \\ 0 & \text{otherwise.} \end{cases} \tag{12}$$

Leveraging the well-designed gradient scaling matrix $\boldsymbol{Q}_k$ in (11), we can efficiently compute the approximate Newton direction, as demonstrated in Proposition 3.3, with proof in Appendix B.2.

**Proposition 3.3.** *If $\boldsymbol{M}_k$ is constructed by* (11)*, then the search direction $\boldsymbol{P}_k$ defined in* (6) *over $\mathcal{I}_k^c$ can be written as $[\boldsymbol{P}_k]_{\mathcal{I}_k^c} = [\boldsymbol{X}_k \mathcal{P}_{\mathcal{I}_k^c}(\nabla f(\boldsymbol{X}_k))\boldsymbol{X}_k]_{\mathcal{I}_k^c}$.*

Using the search direction from Proposition 3.3, we update $\boldsymbol{X}_{k+1}$ over $\mathcal{I}_k^c$ as follows,

$$[\boldsymbol{X}_{k+1}]_{\mathcal{I}_k^c} = \mathcal{P}_\Omega\Big([\boldsymbol{X}_k]_{\mathcal{I}_k^c} - \gamma_k[\boldsymbol{X}_k \mathcal{P}_{\mathcal{I}_k^c}(\nabla f(\boldsymbol{X}_k))\boldsymbol{X}_k]_{\mathcal{I}_k^c}\Big). \tag{13}$$

For the *restricted* variables in the set $\mathcal{I}_k$, we directly set them to zero, *i.e.*, $[\boldsymbol{X}_{k+1}]_{\mathcal{I}_k} = \boldsymbol{0}$.

### 3.3 Computing step size

We adopt an Armijo-like rule for step size selection, ensuring the global convergence of our algorithm. Based on the iterate proposed in Section 3.2, we define $\boldsymbol{X}_k(\gamma_k)$ with $[\boldsymbol{X}_k(\gamma_k)]_{\mathcal{I}_k} = \boldsymbol{0}$ and

$$[\boldsymbol{X}_k(\gamma_k)]_{\mathcal{I}_k^c} = \mathcal{P}_\Omega\Big([\boldsymbol{X}_k]_{\mathcal{I}_k^c} - \gamma_k[\boldsymbol{X}_k \mathcal{P}_{\mathcal{I}_k^c}(\nabla f(\boldsymbol{X}_k))\boldsymbol{X}_k]_{\mathcal{I}_k^c}\Big). \tag{14}$$

We test step sizes $\gamma_k \in \{\beta^0, \beta^1, \beta^2, \dots\}$ with $\beta \in (0,1)$, until we find the smallest $t \in \mathbb{N}$ such that $\boldsymbol{X}_k(\gamma_k)$, with $\gamma_k = \beta^t$, satisfies $\boldsymbol{X}_k(\gamma_k) \in \mathbb{S}_{++}^p$ and the line search condition:

$$f(\boldsymbol{X}_k(\gamma_k)) \leq f(\boldsymbol{X}_k) - \alpha\gamma_k\big\langle[\nabla f(\boldsymbol{X}_k)]_{\mathcal{I}_k^c}, [\boldsymbol{P}_k]_{\mathcal{I}_k^c}\big\rangle - \alpha\big\langle[\nabla f(\boldsymbol{X}_k)]_{\mathcal{I}_k}, [\boldsymbol{X}_k]_{\mathcal{I}_k}\big\rangle, \tag{15}$$

where $\alpha \in (0,1)$ is a scalar. We then set $\boldsymbol{X}_{k+1} = \boldsymbol{X}_k(\gamma_k)$. Positive definiteness of $\boldsymbol{X}_{k+1}$ can be verified during Cholesky factorization for objective function evaluation. It is worth mentioning that working with the positive semi-definiteness constraint on $\boldsymbol{X}$ instead of positive definiteness would not change anything in the algorithm if we keep the line search, as the positive definiteness is automatically enforced due to the form of the objective function.

The line search condition (15) is a variant of the Armijo rule. Condition (15) can be always satisfied for a small enough step size as shown in Proposition 3.4. Define the feasible set of Problem (1) as

$$\mathcal{U}^p := \big\{\boldsymbol{X} \in \mathbb{R}^{p \times p} \,|\, \boldsymbol{X} \in \mathcal{M}^p, \, X_{ij} = 0, \, \forall\, (i,j) \in \mathcal{E}\big\}. \tag{16}$$

For any given $\boldsymbol{X}^o \in \mathcal{U}^p$, define the lower level set of the objective function $f$ for Problem (1) as:

$$L_f := \{\boldsymbol{X} \in \mathcal{U}^p \,|\, f(\boldsymbol{X}) \leq f(\boldsymbol{X}^o)\}. \tag{17}$$

**Proposition 3.4.** *For any $\boldsymbol{X}_k \in L_f$, there exists a $\bar{\gamma}_k > 0$ such that $\boldsymbol{X}_k(\gamma_k) \in \mathbb{S}_{++}^p$ and the line search condition* (15) *holds for any $\gamma_k \in (0, \bar{\gamma}_k)$.*

The proof of Proposition 3.4 is available in Appendix B.3. We demonstrate that $\boldsymbol{X}_k(\gamma_k)$ ensures a decrease of the objective function value in Proposition 3.5, proved in Appendix B.4.

**Proposition 3.5.** *For any $\boldsymbol{X}_k \in L_f$, if $\boldsymbol{X}_k(\gamma_k)$ satisfies the line search condition* (15)*, then we have*

$$f(\boldsymbol{X}_k(\gamma_k)) \leq f(\boldsymbol{X}_k) - \alpha\gamma_k m^2 \big\|[\nabla f(\boldsymbol{X}_k)]_{\mathcal{I}_k^c}\big\|^2,$$

*where $m$ is a positive scalar (See Lemma B.1 in Appendix).*

### 3.4 Computation and memory costs

In each iteration, our algorithm calculates the gradient, performs two matrix multiplications, and conducts two projections, with respective computational costs of $O(p^3)$, $O(p^3)$, and $O(p^2)$. In our current implementation of the line search method, we first conduct the Cholesky factorization $\boldsymbol{X} = \boldsymbol{L}\boldsymbol{L}^\top$ using MATLAB's "chol" function. This function can simultaneously verify the positive definiteness of $\boldsymbol{X}$. Next, we calculate the log-determinant function as $\log\det(\boldsymbol{X}) = 2\sum_i \log(L_{ii})$. The Cholesky factorization is the most computationally demanding step, generally requiring $O(p^3)$ costs for a $p \times p$ matrix.

---

**Algorithm 1** Fast Projected Newton-like (FPN) algorithm

---

1: **Input:** Regularization parameter $\lambda_{ij}$, $\alpha$, $\beta$, $\boldsymbol{S}$, and $\boldsymbol{X}_0$;
2: **for** $k = 0, 1, 2, \ldots$ **do**
3:     Identify the *restricted* set $\mathcal{I}_k$ and *free* set $\mathcal{I}_k^c$ according to (8);
4:     Compute the search direction over $\mathcal{I}_k^c$: $[\boldsymbol{P}_k]_{\mathcal{I}_k^c} = [\boldsymbol{X}_k \mathcal{P}_{\mathcal{I}_k^c}(\nabla f(\boldsymbol{X}_k))\boldsymbol{X}_k]_{\mathcal{I}_k^c}$;
5:     $t \leftarrow 0$;
6:     **repeat**
7:         Update $\boldsymbol{X}_{k+1}$: $[\boldsymbol{X}_{k+1}]_{\mathcal{I}_k} = \boldsymbol{0}$, and $[\boldsymbol{X}_{k+1}]_{\mathcal{I}_k^c} = \mathcal{P}_\Omega\big([\boldsymbol{X}_k]_{\mathcal{I}_k^c} - \beta^t[\boldsymbol{P}_k]_{\mathcal{I}_k^c}\big)$;
8:         $t \leftarrow t + 1$;
9:     **until** $\boldsymbol{X}_{k+1} \succ \boldsymbol{0}$, and

$$f(\boldsymbol{X}_{k+1}) \leq f(\boldsymbol{X}_k) - \alpha\beta^t\langle[\nabla f(\boldsymbol{X}_k)]_{\mathcal{I}_k^c}, [\boldsymbol{P}_k]_{\mathcal{I}_k^c}\rangle - \alpha\langle[\nabla f(\boldsymbol{X}_k)]_{\mathcal{I}_k}, [\boldsymbol{X}_k]_{\mathcal{I}_k}\rangle.$$

10: **end for**

---

To mitigate the computational burden associated with Cholesky factorization, we suggest a more efficient method for evaluating the log-determinant function and verifying positive definiteness, as presented in [38]. This method, which leverages Schur complements and sparse linear system solving, can tackle problems of up to $10^6$ dimensions. Furthermore, it is worthwhile to investigate more efficient strategies for computing an approximate log-determinant function. In this context, the approach proposed in [39] offers a nearly linear scaling of execution time with the number of non-zero entries, while maintaining a high level of accuracy.

In summary, our algorithm has an overall complexity of $O(p^3)$ per iteration. BCD-type algorithms [2, 3, 8] need $O(p^4)$ operations per cycle, while projected quasi-Newton with L-BFGS [36] requires $O((m+p)p^2)$ operations per iteration, with $m$ as the stored iteration count for Hessian approximation. The PPA algorithm [9] requires computing an inexact Newton direction from a $p^2 \times p^2$ linear system during each inner loop iteration, with the exact complexity not established. In addition, our algorithm, gradient projection and BCD-type methods [2, 3, 8] need $O(p^2)$ memory costs, while projected quasi-Newton with L-BFGS [36] requires $O(mp^2)$ and PPA [9] demands $O(p^4)$.

## 4 Convergence Analysis

Prior to delving into the convergence analysis, we first establish the uniqueness of the minimizer for Problem (1) and determine the sufficient and necessary conditions for a point to be the minimizer.

**Theorem 4.1.** *The minimizer of Problem* (1) *is unique, and a point* $\boldsymbol{X}^\star \in \mathcal{M}^p$ *is the minimizer if and only if it satisfies*

$$[\boldsymbol{X}^\star]_{ij} = 0 \ \forall (i,j) \in \mathcal{E}, \quad [\nabla f(\boldsymbol{X}^\star)]_{\mathcal{V}\setminus\mathcal{E}} \leq \boldsymbol{0}, \quad \text{and} \quad [\nabla f(\boldsymbol{X}^\star)]_{\mathcal{V}^c} = \boldsymbol{0}, \tag{18}$$

*where* $\mathcal{V} = \{(i,j) \in [p]^2 \mid [\boldsymbol{X}^\star]_{ij} = 0\}$.

The proof of Theorem 4.1 is available in Appendix B.5. The following theorem shows that our algorithm converges to the minimizer of Problem (1).

**Theorem 4.2.** *The sequence* $\{\boldsymbol{X}_k\}$ *generated by Algorithm 1 with* $\boldsymbol{X}_0 \in L_f$ *converges to the minimizer* $\boldsymbol{X}^\star$ *of Problem* (1)*, with* $\{f(\boldsymbol{X}_k)\}$ *monotonically decreasing.*

The proof of Theorem 4.2 is available in Appendix B.6. It is worth noting that constructing an initial point $\boldsymbol{X}_0 \in L_f$, as defined in (17), is straightforward. Please refer to the proof of Theorem 4.2 for more details on this. The theoretical analysis on support set convergence and sequence convergence rate relies on the following assumption.

**Assumption 4.3.** For the minimizer $\boldsymbol{X}^\star$ of Problem (1), we assume that the gradient of the objective function $f$ at $\boldsymbol{X}^\star$ satisfies

$$\big[\nabla f(\boldsymbol{X}^\star)\big]_{ij} < 0, \quad \forall (i,j) \in \mathcal{V} \setminus \mathcal{E},$$

where $\mathcal{V} = \big\{(i,j) \in [p]^2 \,\big|\, \big[\boldsymbol{X}^\star\big]_{ij} = 0\big\}$, and $\mathcal{E}$ is the disconnectivity set.

**Theorem 4.4.** *Under Assumption 4.3, the set of free variables $\mathcal{I}_k^c$ generated by Algorithm 1 converges to the support of the minimizer $\boldsymbol{X}^\star$ of Problem (1) in finite iterations. In other words, there exists some $k_o \in \mathbb{N}_+$ such that $\mathcal{I}_k^c = \operatorname{supp}(\boldsymbol{X}^\star)$ for any $k \geq k_o$.*

The proof of Theorem 4.4 is provided in Appendix B.7. Theorem 4.4 demonstrates that the set of *free* variables constructed by our variable partitioning scheme can exactly identify the support of $\boldsymbol{X}^\star$ in finite iterations. Now we establish the convergence rate of our algorithm. Define

$$\boldsymbol{R}_k = \big[\boldsymbol{H}_k\big]_{\mathcal{I}_k^c \mathcal{I}_k^c} - \big[\boldsymbol{H}_k\big]_{\mathcal{I}_k^c \mathcal{I}_k}\big[\boldsymbol{H}_k\big]_{\mathcal{I}_k \mathcal{I}_k}^{-1}\big[\boldsymbol{H}_k\big]_{\mathcal{I}_k^c \mathcal{I}_k}^\top. \tag{19}$$

**Theorem 4.5.** *Under Assumption 4.3, the sequence $\{\boldsymbol{X}_k\}$ generated by Algorithm 1 satisfies*

$$\limsup_{k\to\infty}\frac{\big\|\boldsymbol{X}_{k+1} - \boldsymbol{X}^\star\big\|_{\boldsymbol{M}_k}^2}{\big\|\boldsymbol{X}_k - \boldsymbol{X}^\star\big\|_{\boldsymbol{M}_k}^2} \leq \limsup_{k\to\infty}\left(1 - \min\left(m_k, \frac{2(1-\alpha)\beta m_k}{M_k}\right)\right)^2,$$

*where $m_k$ and $M_k$ as the smallest and largest eigenvalues of $\boldsymbol{R}_k^{-\frac{1}{2}}\big[\boldsymbol{H}_k\big]_{\mathcal{I}_k^c \mathcal{I}_k^c}\boldsymbol{R}_k^{-\frac{1}{2}}$, respectively.*

The proof of Theorem 4.5 is provided in Appendix B.8. Theorem 4.5 reveals that the convergence rate of our algorithm depends on the condition number $m_k/M_k$ of $\boldsymbol{R}_k^{-\frac{1}{2}}[\boldsymbol{H}_k]_{\mathcal{I}_k^c \mathcal{I}_k^c}\boldsymbol{R}_k^{-\frac{1}{2}}$. Replacing $\boldsymbol{R}_k$ with an identify matrix, (*i.e.*, using the projected gradient method) results in a rate dependent on the condition number of $[\boldsymbol{H}_k]_{\mathcal{I}_k^c \mathcal{I}_k^c}$. The condition number of $\boldsymbol{R}_k^{-\frac{1}{2}}[\boldsymbol{H}_k]_{\mathcal{I}_k^c \mathcal{I}_k^c}\boldsymbol{R}_k^{-\frac{1}{2}}$ could be larger, as $\boldsymbol{R}_k$ could approximate $[\boldsymbol{H}_k]_{\mathcal{I}_k^c \mathcal{I}_k^c}$ well. Thus, the gradient scaling matrix $\boldsymbol{R}_k^{-1}$, *i.e.*, $[\boldsymbol{H}_k^{-1}]_{\mathcal{I}_k^c \mathcal{I}_k^c}$ in (11), leads our algorithm to converge faster than the projected gradient method.

It is important to note that our algorithm generally does not achieve superlinear convergence, despite the incorporation of second-order information. Superlinear convergence necessitates that the inverse gradient scaling matrix progressively approximate the Hessian at the minimizer [40]. However, this is a condition that our constructed scaling matrix does not meet. Despite this, we should note that constructing a search direction to achieve superlinear convergence proves significantly more computationally demanding than our approach, as it cannot leverage the special structure of the Hessian to decrease the computational load.

## 5    Experimental Results

We conduct experiments on synthetic and real-world data to verify the performance of our algorithm. All experiments were conducted on 2.10GHZ Xeon Gold 6152 machines and Linux OS, and all methods were implemented in MATLAB. State-of-the-art methods for comparisons include:

- BCD [2]: Updates each column/row of primal variable using a nonnegative quadratic program.
- optGL [8]: Similar to BCD but solves nonnegative quadratic programs on the dual variable.
- GGL [3]: Similar to BCD but handles disconnectivity constraints, while BCD and optGL cannot.
- PGD [13]: Projected gradient descent method with backtracking line search.
- APGD [41]: Accelerated projected gradient algorithm with extrapolation step.
- PPA [9]: Inexact proximal point algorithm with Newton-CG method.
- PQN-LBFGS[36]: Projected quasi-Newton method using limited-memory BFGS.

Note that all state-of-the-art methods listed above can converge to the minimizer of Problem (1), and we focus on the comparison of computational time for those methods. To that end, we report the relative error of the objective function value as a function of the run time, which is calculated by

$$|f(\boldsymbol{X}_k) - f(\boldsymbol{X}^\star)|/|f(\boldsymbol{X}^\star)|, \tag{20}$$

where $f$ is the objective function of Problem (1), and $\boldsymbol{X}^\star$ is its minimizer. The $\boldsymbol{X}^\star$ is computed by running the state-of-the-art method GGL [3] until it converges to a point $\boldsymbol{X}_k \in \mathcal{M}^p$ satisfying

$$[\boldsymbol{X}_k]_{ij} = 0 \quad \forall\,(i,j) \in \mathcal{E}, \qquad [\nabla f(\boldsymbol{X}_k)]_{\mathcal{A}\backslash\mathcal{E}} \leq \boldsymbol{0}, \qquad \|[\nabla f(\boldsymbol{X}^\star)]_{\mathcal{A}^c}\|_{\max} \leq 10^{-8}, \tag{21}$$

where $\mathcal{A} := \{(i,j) \in [p]^2 \,\big|\, |[\boldsymbol{X}_k]_{ij}| \leq 10^{-8}\}$. Through the comparison with the sufficient and necessary conditions of the unique minimizer of Problem (1) presented in Theorem 4.1, we can see that any point $\boldsymbol{X}_k$ satisfying the conditions in (21) is sufficiently close to the minimizer.

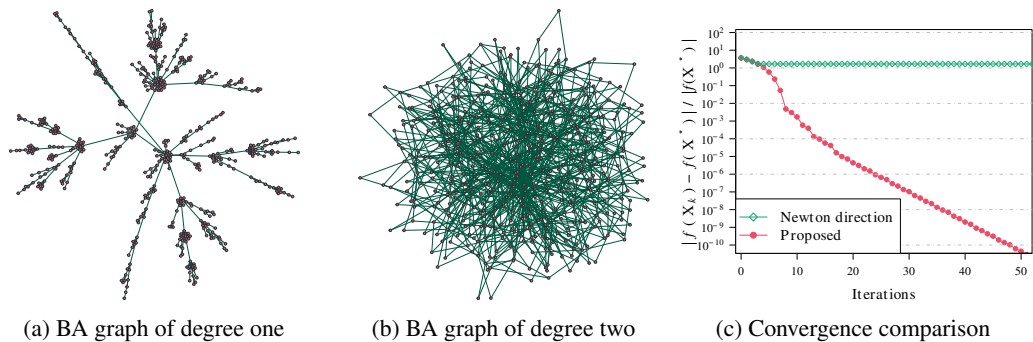

| (a) BA graph of degree one | (b) BA graph of degree two | (c) Convergence comparison |

Figure 1: Illustration of BA graphs of degree one (a) and degree two (b) with 500 nodes. (c) Convergence comparison between our algorithm and the Newton direction from Proposition 3.1.

## 5.1 Synthetic data

We generate independent samples $\boldsymbol{y}^{(1)}, \ldots, \boldsymbol{y}^{(n)} \in \mathbb{R}^p$ from a multivariate Gaussian distribution with zero mean and precision matrix $\boldsymbol{\Theta}$, where $\boldsymbol{\Theta} \in \mathcal{M}^p$ is the underlying precision matrix associated with a graph consisting of $p$ nodes. We use the Barabási–Albert (BA) model [42] to generate the support of the underlying precision matrix. To help readers to know well the BA graphs, we present two examples in Figure 1. More details about experimental setting are provided in Appendix A.

### 5.1.1 Comparisons of search directions

We evaluate the convergence of algorithms with different search directions. Figure 1 (c) demonstrates that our algorithm, using the direction from Proposition 3.3, converges to the minimizer, aligning with our theoretical convergence results in Theorem 4.2. In contrast, the algorithm using iterate (5) and the Newton direction from Proposition 3.1 stops decreasing the objective function value after a few iterations, indicating that this direction cannot be consistently regarded as a descent direction. The algorithm selects the step size through the Armijo rule, *i.e.*, it is continually reduced until a decrease in the objective function is achieved.

### 5.1.2 Comparisons of computational time

We evaluate the computational time of our algorithm and state-of-the-art methods on synthetic datasets, averaging results over 10 realizations. We plot markers every 10 iterations for PGD, APGD, PQN-LBFGS, and FPN, while marking each cycle of updating all columns/rows for BCD-type algorithms (BCD, GGL, and optGL) and each outer iteration for PPA.

Figure 2 compares the computational time of various methods for solving Problem (1) on BA graphs of degree one. Our proposed FPN significantly outperforms all state-of-the-art methods in convergence time for node counts ranging from 1000 to 5000. BCD and GGL are efficient at 1000 nodes, being faster than PGD and APGD, and competitive with PQN-LBFGS and PPA. However, at 5000 nodes, BCD and GGL become slower due to the $O(p^4)$ operations per cycle required to solve $p$ nonnegative quadratic programs, leading to rapidly increasing computational costs.

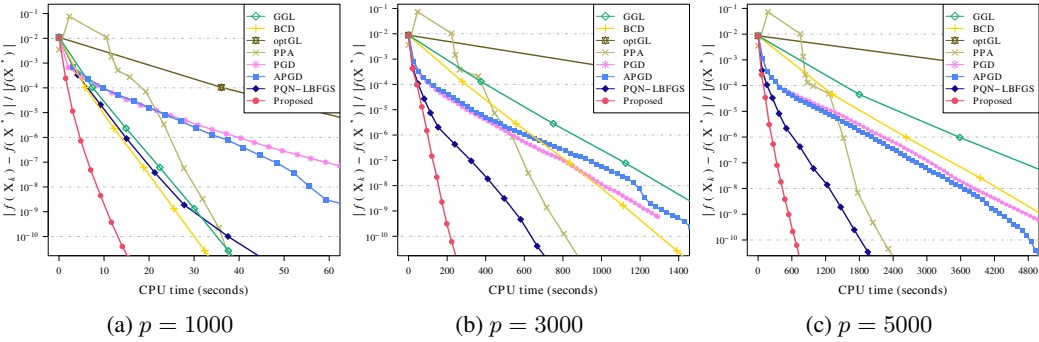

| (a) $p = 1000$ | (b) $p = 3000$ | (c) $p = 5000$ |

Figure 2: Relative errors of the objective function values versus time on BA graphs of degree one.

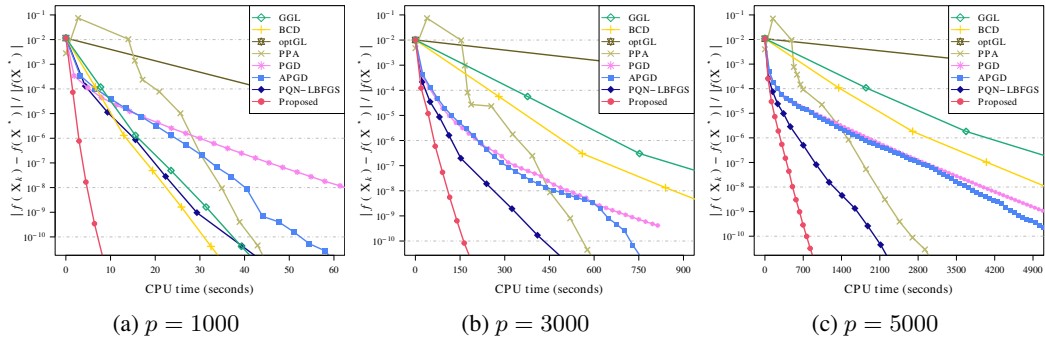

(a) $p = 1000$    (b) $p = 3000$    (c) $p = 5000$

Figure 3: Relative errors of the objective function value versus time on BA graphs of degree two.

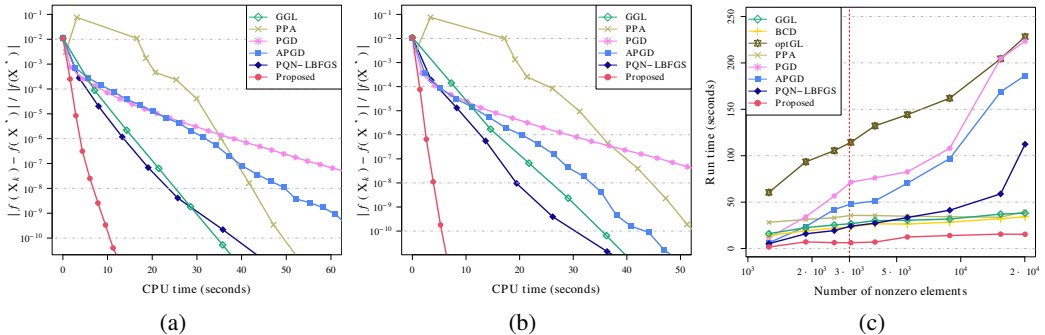

(a)    (b)    (c)

Figure 4: Relative errors of objective function values versus time on data sets: (a) BA graph of degree one and (b) BA graph of degree two, imposing disconnectivity constraints. (c) Run time versus numbers of nonzero elements in precision matrices across varying regularization parameter values, with the underlying precision matrix having 2998 nonzero elements (indicated by vertical red line).

Figure 3 presents the computational time of different methods on BA graphs of degree two. As with degree one graphs, FPN outperforms state-of-the-art methods in computational time for varying node counts. PQN-LBFGS and FPN require fewer iterations to converge to the minimizer than PGD and APGD, particularly in high dimensions (*e.g.*, $p = 5000$), indicating faster convergence. This is because both PQN-LBFGS and FPN utilize second-order information and approximate Newton direction, overcoming low convergence rates of first-order methods in high-dimensional cases.

Figure 4 (a) and (b) compare the computational time of various algorithms solving Problem (1) with disconnectivity constraints, where FPN consistently converges fastest. (c) evaluates the impact of the estimated precision matrix's sparsity level on run time. BCD, GGL, PPA, and FPN exhibit stable run time across varying sparsity levels, highlighting their robustness regarding regularization parameter settings, while other methods display increased run time as sparsity decreases.

## 5.2   Real-world data

We perform experiments on two real-world datasets: the *concepts* dataset and a financial time-series dataset. For the *concepts* dataset, we compare the computational time of different algorithms solving Problem (1). The experimental results on the financial time-series dataset are provided in Appendix A, where we examine the performance of our method in graph edge recovery.

The *concepts* dataset [43], from Intel Labs, comprises 1000 nodes and 218 semantic features, with $p = 1000$ and $n = 218$. Nodes represent concepts like "house," "coat," and "whale," while semantic features are questions like "Can it fly?", "Is it alive?", and "Can you use it?". Responses, collected via Amazon Mechanical Turk, range from "definitely no" to "definitely yes" on a five-point scale.

Figure 5 (a) compares the run time of various algorithms solving Problem (1) on the *concepts* dataset. Our proposed algorithm converges to the minimizer considerably faster than state-of-the-art algorithms, which is consistent with the observations in synthetic experiments. Note that all compared algorithms can reach the minimizer of Problem (1), and thus learn the same graph.

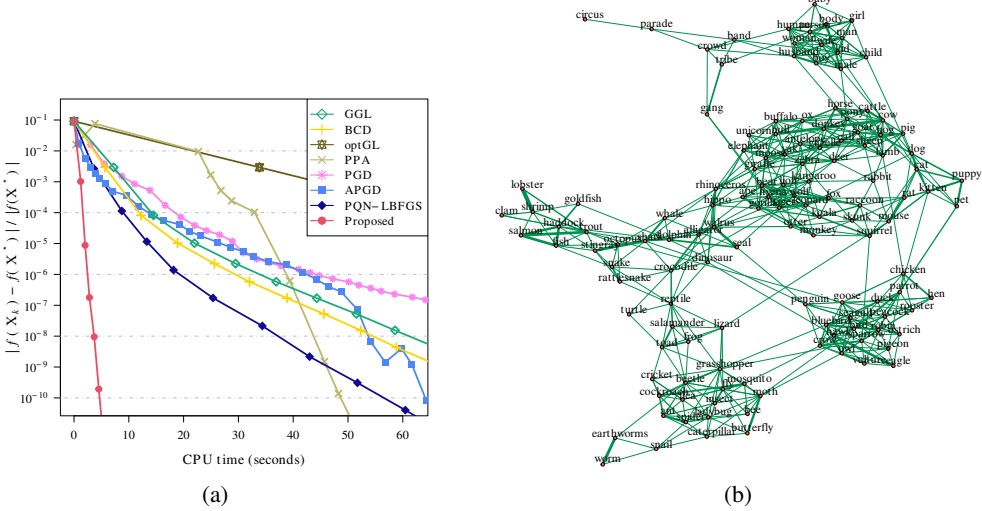

(a)                                                    (b)

Figure 5: (a) Relative errors of the objective function values versus time for different algorithms in solving Problem (1) on the *concepts* data set, consisting of 1000 nodes. (b) The connected subgraph illustrated by the minimizer on the *concepts* data set, which consists of 132 nodes.

Figure 5 (b) displays a connected subgraph illustrated by the minimizer of Problem (1). Interestingly, it is observed that the learned graph forms a semantic network, where related concepts are closely connected. For instance, insect concepts such as "bee", "butterfly", "flea", "mosquito", and "spider" are grouped together, while human-related concepts like "baby", "husband", "child", "girls", and "man" form another group. Moreover, the network connects "penguin" closely to birds like "owl' and "crow" and sea animals like "goldfish" and "seal", highlighting its aquatic bird nature. Overall, the learned network effectively captures concept relationships.

## 6  Conclusions and Discussions

In this paper, we have introduced a fast projected Newton-like method for estimating precision matrices under $\mathrm{MTP}_2$ constraints. Our algorithm, leveraging the two-metric projection method, stands out from existing BCD and PPA-type approaches for addressing the target problem. The proposed algorithm is not only straightforward to implement but also efficient in terms of computation and memory usage. We have provided theoretical convergence analysis and conducted extensive experiments, which clearly demonstrate the superior efficiency of our algorithm in computational time, outperforming state-of-the-art methods. Moreover, we have observed significant performance of our method in terms of *modularity* value on the learned financial time-series graphs.

Finally, we discuss the limitations of our paper. Our algorithm is proven to converge to the minimizer without any assumptions; however, we require Assumption 4.3 to establish support set convergence in finite iterations and to determine the convergence rate. This assumption is relatively mild, as Theorem 4.1 shows that the minimizer $\boldsymbol{X}^\star$ must satisfy $[\nabla f(\boldsymbol{X}^\star)]_{ij} \leq 0$ for each $(i,j) \in \mathcal{V}\backslash\mathcal{E}$. The only additional requirement in Assumption 4.3 is the strictness of this inequality. However, the conditions for ensuring this strictness remain unclear. As this assumption is equivalent to the strict complementary slackness condition in optimization theory, exploring verifiable conditions to guarantee Assumption 4.3 could enrich our algorithm's insights.

## 7  Acknowledgements

This work was supported by the Hong Kong Research Grants Council GRF 16207820, 16310620, and 16306821, the Hong Kong Innovation and Technology Fund (ITF) MHP/009/20, and the Project of Hetao Shenzhen-Hong Kong Science and Technology Innovation Cooperation Zone under Grant HZQB-KCZYB-2020083. We would also like to thank the anonymous reviewers for their valuable feedback on the manuscript.

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

In the following sections, we provide more details on experimental settings and additional experimental results for financial time-series data in Appendix A and the proofs for Propositions 3.1, 3.3, 3.4, and 3.5, as well as Theorems 4.1, 4.2, 4.4, and 4.5 in Appendix B.

## A    Additional Experiments

We first provide an in-depth description of the experimental settings, then present numerical experiments carried out on the financial time-series data.

### A.1    Experimental settings

We examine Barabasi-Albert (BA) graphs [42] as a model for the support structure of the underlying precision matrix. BA models hold a significant position in network science due to their ability to generate random scale-free networks via a *preferential attachment* mechanism. This mechanism ensures that newly introduced nodes are more likely to connect with nodes that possess a higher degree during the network's evolution. Scale-free networks serve as suitable models for various systems, including the Internet, protein interaction networks, citation networks, as well as the majority of social and online networks [42].

In a BA graph with degree $r$, each new node connects to $r$ pre-existing nodes, with the probability of connection being proportional to the number of edges the existing nodes currently have. In this paper, we consider $r$ values of 1 and 2.

Following the procedures detailed in [44], we assign a positive weight to each edge of a graph and set a zero weight for disconnected nodes. Positive weights are uniformly sampled from $U(2, 5)$. This process results in a weighted adjacency matrix $\boldsymbol{A}$ containing all the graph weights. Then we adopt the procedures outlined in [2] for generating the underlying precision matrix $\boldsymbol{\Theta}$. We first set

$$\tilde{\boldsymbol{\Theta}} = \delta \boldsymbol{I} - \boldsymbol{A}, \quad \text{with} \quad \delta = 1.05 \lambda_{\max}(\boldsymbol{A}), \tag{22}$$

where $\lambda_{\max}(\boldsymbol{A})$ represents the largest eigenvalue of $\boldsymbol{A}$. In this context, the weight is zero if two nodes are disconnected, and follows a uniform distribution $U(2, 5)$ when nodes are connected. Lastly, we define $\boldsymbol{\Theta} = \boldsymbol{D}\tilde{\boldsymbol{\Theta}}\boldsymbol{D}$, where $\boldsymbol{D}$ is a diagonal matrix selected such that the covariance matrix $\boldsymbol{\Theta}^{-1}$ has unit diagonal elements.

We set the regularization parameter $\lambda_{ij}$ in Problem (1) as follows

$$\lambda_{ij} = \frac{\sigma}{\left|[\hat{\boldsymbol{X}}]_{ij}\right| + \epsilon}, \quad \forall i \neq j, \tag{23}$$

where $\hat{\boldsymbol{X}}$ represents an estimator, $\epsilon$ is set to $10^{-3}$, and $\sigma > 0$ is a parameter that adjusts the sparsity. The function in (23) is closely related to the re-weighted $\ell_1$-norm regularization [14, 45, 46], which effectively enhances the sparsity of the solution and reduces the estimation bias resulting from the $\ell_1$-norm [47]. We employ the maximum likelihood estimator as $\hat{\boldsymbol{X}}$, *i.e.*, the minimizer of Problem (1) without the sparsity regularization. Note that one can solve the maximum likelihood estimator with a relatively large tolerance to obtain a coarse estimator, and alternative monotonically decreasing functions may be explored in (23). For PQN-LBFGS, we utilize the previous 50 updates to compute the search direction. For FPN, we set $\epsilon_k = 10^{-15}$ in (8) for identifying the set of *restricted* variables.

For calculating the relative error as defined in (20), there is no additional computational cost for PGD, APGD, PQN-LBFGS and FPN, since the objective function value is already evaluated within the backtracking line search during each iteration. In contrast, BCD, optGL, and GGL do not require evaluating the objective function, leading to an extra computational cost when computing the relative error in (20) for comparison purposes. However, this additional cost can be considered negligible, as these methods only compute the objective function value after completing a cycle. Moreover, the number of cycles needed for BCD, optGL, and GGL is substantially fewer than the number of iterations required by other methods.

## A.2 Financial time-series data

We carry out numerical experiments on a financial time-series dataset to evaluate the performance of our method in recovering graph edges. The applicability of MTP2 models on financial time-series data is well-established, as market factors lead to positive dependencies among stocks [5].

The dataset consists of 201 stocks composing the S&P 500 index, spanning the period from January 1, 2017, to January 1, 2020, yielding 753 observations per stock, *i.e.*, $p = 201$ and $n = 753$. We construct the log-returns data matrix as $Xi, j = \log P_{i,j} - \log P_{i-1,j}$, where $P_{i,j}$ represents the closing price of the $j$-th stock on the $i$-th day. The stocks are categorized into five sectors based on the Global Industry Classification Standard (GICS) system: Consumer Staples, Utilities, Industrials, Information Technology, and Energy.

Directly assessing the correctness of the learned graph edges is not feasible in financial time-series data, as the underlying graph structure remains unknown. However, we expect stocks from the same sector to have interconnected edges. To measure the performance of edge recovery, we employ the *modularity* metric [48]. Given a graph $\mathcal{G} = (V, E)$, where $V$ represents the vertex set and $E$ denotes the edge set, the *modularity* is defined as:

$$Q := \frac{1}{2|E|} \sum_{i,j \in V} \left( A_{ij} - \frac{d_i d_j}{2|E|} \right) \delta(c_i, c_j), \tag{24}$$

where $A_{ij} = 1$ if $(i, j) \in E$, and 0 otherwise. $d_i$ represents the number of edges connected to node $i$, $c_i$ indicates the type of node $i$, and $\delta(\cdot, \cdot)$ refers to the Kronecker delta function, with $\delta(a, b) = 1$ if $a = b$ and 0 otherwise. A stock graph with high modularity exhibits dense connections among stocks within the same sector and sparse connections between stocks in distinct sectors. A higher *modularity* value implies a more faithful representation of the underlying stock network.

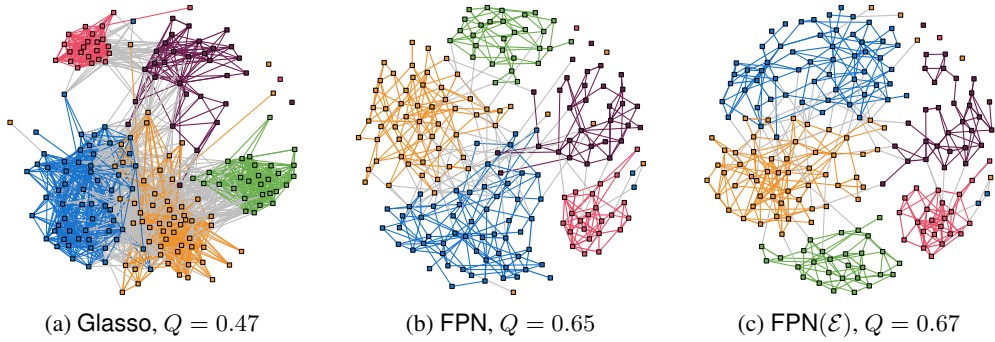

| (a) Glasso, $Q = 0.47$ | (b) FPN, $Q = 0.65$ | (c) FPN($\mathcal{E}$), $Q = 0.67$ |

Figure 6: Financial time-series graphs learned via (a) Glasso, (b) FPN, and (c) FPN($\mathcal{E}$).

Figure 6 demonstrates that the performances of our proposed FPN and FPN($\mathcal{E}$) are better than that of Glasso [22], since the majority of connections in graphs learned through FPN and FPN($\mathcal{E}$) occur between nodes within the same sector. In contrast, only a few connections (depicted as gray-colored edges) exist between nodes from different sectors. Both FPN and FPN($\mathcal{E}$) achieve higher *modularity* values compared to Glasso, indicating that the former have a higher degree of interpretability than the latter. Furthermore, we observe that FPN($\mathcal{E}$) moderately enhances the performance of FPN.

We fine-tune the sparsity regularization parameter for each method based on the *modularity* value, allowing only a limited number of isolated nodes. Note that increasing the regularization parameter for Glasso would result in numerous isolated nodes that cannot be grouped. FPN($\mathcal{E}$) refers to the application of FPN for solving Problem (1) with a disconnectivity set $\mathcal{E}$. This set is obtained through hard thresholding on the MLE, which is also used in computing regularization weights in (23).

# B Proofs

In this section, we provide the proofs for Propositions 3.1, 3.3, 3.4, and 3.5, as well as Theorems 4.1, 4.2, 4.4, and 4.5.

## B.1 Proof of Proposition 3.1

*Proof.* The Hessian matrix at $\boldsymbol{X}_k$ has the form:

$$\boldsymbol{H}_k = \boldsymbol{X}_k^{-1} \otimes \boldsymbol{X}_k^{-1}.$$

Then we obtain

$$\boldsymbol{H}_k^{-1} = \boldsymbol{X}_k \otimes \boldsymbol{X}_k.$$

Following from the property of the Kronecker product that $\mathrm{vec}\left(\boldsymbol{ABC}\right) = \left(\boldsymbol{C}^\top \otimes \boldsymbol{A}\right)\mathrm{vec}(\boldsymbol{B})$, we can obtain

$$\mathrm{vec}\left(\boldsymbol{P}_k\right) = \boldsymbol{H}_k^{-1}\mathrm{vec}\left(\nabla f(\boldsymbol{X}_k)\right) = \mathrm{vec}\left(\boldsymbol{X}_k \nabla f(\boldsymbol{X}_k)\boldsymbol{X}_k\right).$$

As a result, we have

$$\boldsymbol{P}_k = \boldsymbol{X}_k \nabla f(\boldsymbol{X}_k)\boldsymbol{X}_k,$$

completing the proof. $\qquad\square$

## B.2 Proof of Proposition 3.3

*Proof.* Following from (10) and (11), we obtain

$$\left[\boldsymbol{P}_k\right]_{\mathcal{I}_k^c} = \left[\boldsymbol{H}_k^{-1}\right]_{\mathcal{I}_k^c \mathcal{I}_k^c}\left[\nabla f(\boldsymbol{X}_k)\right]_{\mathcal{I}_k^c} = \left[\boldsymbol{H}_k^{-1}\mathrm{vec}\left(\mathcal{P}_{\mathcal{I}_k^c}\left(\nabla f(\boldsymbol{X}_k)\right)\right)\right]_{\mathcal{I}_k^c},$$

where projection $\mathcal{P}_{\mathcal{I}_k^c}$ is defined in (12). For ease of presentation, by a slight abuse of notation, both $[\boldsymbol{A}]_{\mathcal{I}_k^c} \in \mathbb{R}^{|\mathcal{I}_k^c|}$ and $[\mathrm{vec}\left(\boldsymbol{A}\right)]_{\mathcal{I}_k^c} \in \mathbb{R}^{|\mathcal{I}_k^c|}$ represent a vector containing all elements of $\boldsymbol{A}$ in the set $\mathcal{I}_k^c$. Following from the fact that $\mathrm{vec}\left(\boldsymbol{ABC}\right) = \left(\boldsymbol{C}^\top \otimes \boldsymbol{A}\right)\mathrm{vec}(\boldsymbol{B})$ and $\boldsymbol{H}_k^{-1} = \boldsymbol{X}_k \otimes \boldsymbol{X}_k$, we have

$$\boldsymbol{H}_k^{-1}\mathrm{vec}\left(\mathcal{P}_{\mathcal{I}_k^c}\left(\nabla f(\boldsymbol{X}_k)\right)\right) = \mathrm{vec}\left(\boldsymbol{X}_k \mathcal{P}_{\mathcal{I}_k^c}\left(\nabla f\left(\boldsymbol{X}_k\right)\right)\boldsymbol{X}_k\right).$$

By collecting the elements in the set $\mathcal{I}_k^c$, we obtain

$$\left[\boldsymbol{P}_k\right]_{\mathcal{I}_k^c} = \left[\boldsymbol{X}_k \mathcal{P}_{\mathcal{I}_k^c}\left(\nabla f\left(\boldsymbol{X}_k\right)\right)\boldsymbol{X}_k\right]_{\mathcal{I}_k^c},$$

completing the proof. $\qquad\square$

## B.3 Proof of Propositio 3.4

To prove Propositio 3.4, we first establish Lemma B.1 below to show that the lower level set of the objective function is compact. We note that Lemma B.1 depends on the condition that the sample covariance matrix has strictly positive diagonal elements, which is assumed throughout the paper and holds with probability one.

**Lemma B.1.** *The lower level set $L_f$ defined in (17) is nonempty and compact, and for any $\boldsymbol{X} \in L_f$, we have*

$$m\boldsymbol{I} \preceq \boldsymbol{X} \preceq M\boldsymbol{I},$$

*where $m$ and $M$ are two positive scalars.*

*Proof.* We first show that the largest eigenvalue of any $\boldsymbol{X} \in L_f$ can be upper bounded by $M$. The objective function of (1) can be written as

$$f(\boldsymbol{X}) = -\log\det(\boldsymbol{X}) + \mathrm{tr}\left(\boldsymbol{XG}\right),$$

where $\boldsymbol{G} = \boldsymbol{S} - \boldsymbol{\lambda}$ with $\boldsymbol{\lambda}$ defined by

$$[\boldsymbol{\lambda}]_{ij} = \begin{cases} \lambda_{ij} & \text{if } i \neq j, \\ 0 & \text{otherwise.} \end{cases} \tag{25}$$

Define two constants
$$\mu := \min_i G_{ii}, \quad \text{and} \quad \upsilon := \max_{i \neq j, \, (i,j) \notin \mathcal{E}} |G_{ij}|.$$

Note that $\mu > 0$ because $S_{ii} > 0$ for each $i \in [p]$. Then one has

$$\text{tr}\,(\boldsymbol{X}\boldsymbol{G}) = \sum_i G_{ii} X_{ii} + \sum_{i \neq j} G_{ij} X_{ij} \geq \mu \text{tr}\,(\boldsymbol{X}) + \upsilon \sum_{i \neq j} X_{ij}, \tag{26}$$

where the inequality follows from the fact that $X_{ij} \leq 0$ for $i \neq j$, and $X_{ij} = 0$ for $(i,j) \in \mathcal{E}$. We denote the largest eigenvalue of $\boldsymbol{X}$ by $\lambda_{\max}(\boldsymbol{X})$. Then we have

$$\lambda_{\max}(\boldsymbol{X}) \leq \text{tr}\,(\boldsymbol{X}) \leq \frac{1}{\mu}\Big(\text{tr}\,(\boldsymbol{X}\boldsymbol{G}) + \upsilon \sum_{i \neq j} |X_{ij}|\Big), \tag{27}$$

where the second inequality follows from (26). In what follows, we bound the terms $\text{tr}\,(\boldsymbol{X}\boldsymbol{G})$ and $\sum_{i \neq j} X_{ij}$. For any $\boldsymbol{X} \in L_f$, one has $f(\boldsymbol{X}^o) \geq -\log \det(\boldsymbol{X}) + \text{tr}\,(\boldsymbol{X}\boldsymbol{G})$. Therefore, the term $\text{tr}\,(\boldsymbol{X}\boldsymbol{G})$ can be bounded by

$$\text{tr}\,(\boldsymbol{X}\boldsymbol{G}) \leq f(\boldsymbol{X}^o) + \log \det(\boldsymbol{X}) \leq f(\boldsymbol{X}^o) + p \log\left(\lambda_{\max}(\boldsymbol{X})\right). \tag{28}$$

Let $\lambda' := \min_{i \neq j, \, (i,j) \notin \mathcal{E}} \lambda_{ij}$. Then one has

$$\sum_{i \neq j} |X_{ij}| \leq \frac{1}{\lambda'} \sum_{i \neq j} \lambda_{ij} |X_{ij}| \leq \frac{1}{\lambda'}\left(f(\boldsymbol{X}^o) + p \log\left(\lambda_{\max}(\boldsymbol{X})\right)\right), \tag{29}$$

where the last inequality follows from $\sum_{i \neq j} \lambda_{ij} |X_{ij}| \leq \text{tr}\,(\boldsymbol{X}\boldsymbol{G})$, because $\text{tr}\,(\boldsymbol{X}\boldsymbol{S}) \geq 0$ since $\boldsymbol{X} \in \mathbb{S}_{++}^p$ and $\boldsymbol{S} \in \mathbb{S}_+^p$. Together with (27), (28) and (29), we obtain

$$\lambda_{\max}(\boldsymbol{X}) \leq \frac{1}{\mu}\Big(1 + \frac{\upsilon}{\lambda'}\Big)\left(f(\boldsymbol{X}^o) + p \log\left(\lambda_{\max}(\boldsymbol{X})\right)\right).$$

Since $\log\left(\lambda_{\max}(\boldsymbol{X})\right)$ grows much slower than $\lambda_{\max}(\boldsymbol{X})$, $\lambda_{\max}(\boldsymbol{X})$ can be upper bounded by a constant $M$, which depends on $f(\boldsymbol{X}^o)$, $\boldsymbol{S}$, and $\boldsymbol{\lambda}$.

We denote the smallest eigenvalue of $\boldsymbol{X}$ by $\lambda_{\min}(\boldsymbol{X})$. For any $\boldsymbol{X} \in L_f$, one has

$$f(\boldsymbol{X}^o) \geq f(\boldsymbol{X}) \geq -\log \det(\boldsymbol{X}) \geq -\log \lambda_{\min}(\boldsymbol{X}) - (p-1) \log M.$$

As a result, we have $\lambda_{\min}(\boldsymbol{X}) \geq e^{-f(\boldsymbol{X}^o)} M^{-(p-1)}$, which shows that $\lambda_{\min}(\boldsymbol{X})$ can be lower bounded by a positive constant $m = e^{-f(\boldsymbol{X}^o)} M^{-(p-1)}$. Finally, we show that the lower level set $L_f$ is compact. First, $L_f$ is closed because $f$ is a continuous function. Second, $L_f$ is also bounded because it is a subset of $\{\boldsymbol{X} \in \mathbb{R}^{p \times p} | m\boldsymbol{I} \preceq \boldsymbol{X} \preceq M\boldsymbol{I}\}$. $\qquad \square$

*Proof of Propositio 3.4.* We first show that $\boldsymbol{X}_k(\gamma_k) \in \mathbb{S}_{++}$ holds for a small enough step size. The $\boldsymbol{X}_k(\gamma_k)$ can be equivalently written in the form $\boldsymbol{X}_k(\gamma_k) = \boldsymbol{Z}_k - \gamma_k \boldsymbol{G}_k$, where $\boldsymbol{Z}_k \in \mathbb{R}^{p \times p}$ defined by $[\boldsymbol{Z}_k]_{ij} = [\boldsymbol{X}_k]_{ij}$ if $(i,j) \in \mathcal{I}_k^c$, and 0 otherwise. $\boldsymbol{G}_k$ is some search direction only over $\mathcal{I}_k^c$, which is a symmetric matrix. It is known that a matrix $\boldsymbol{A}$ is a nonsingular *M-matrix* if and only if $\boldsymbol{A}$ is a *Z-matrix* and there exists a vector $\boldsymbol{x} > \boldsymbol{0}$ with $\boldsymbol{A}\boldsymbol{x} > \boldsymbol{0}$ [49]. Following from the fact that $\boldsymbol{X}_k$ is a nonsingular *M-matrix*, there exists $\boldsymbol{x} > \boldsymbol{0}$ such that

$$\boldsymbol{Z}_k\boldsymbol{x} \geq \boldsymbol{X}_k\boldsymbol{x} > 0.$$

Therefore, $\boldsymbol{Z}_k$ is also a nonsingular *M-matrix*, implying that $\boldsymbol{Z}_k \in \mathbb{S}_{++}$. As a result, if $\gamma_k < \lambda_{\min}(\boldsymbol{Z}_k)/\rho(\boldsymbol{G}_k)$, where $\rho(\boldsymbol{G}_k)$ is the spectral radius of $\boldsymbol{G}_k$, then $\boldsymbol{X}_k(\gamma_k) \in \mathbb{S}_{++}$. The definition of $\boldsymbol{X}_k(\gamma_k)$ further guarantees that $\boldsymbol{X}_k(\gamma_k) \in \mathcal{U}^p$. We can verify that $f(\boldsymbol{X}) = +\infty$ for any $\boldsymbol{X} \in \text{cl}(\mathcal{U}^p) \setminus \mathcal{U}^p$, where $\boldsymbol{X}$ is positive semidefinite and singular. Thus, we consider an $\boldsymbol{X}^o \in \mathcal{U}^p$ with $f(\boldsymbol{X}^o)$ is sufficiently large such that $f(\boldsymbol{X}_k(\gamma_k)) \leq f(\boldsymbol{X}^o)$. Therefore, $\boldsymbol{X}_k(\gamma_k) \in L_f$.

Next we prove that the line search condition (15) holds for a small enough step size. Recall that $\mathcal{I}_k^c$ is the complement of the set $\mathcal{I}_k$ defined in (8). For any $\boldsymbol{X}_k \in L_f$, the set $\mathcal{I}_k^c$ can be represented as

$$\mathcal{I}_k^c = \mathcal{T}^c(\boldsymbol{X}_k, \epsilon_k) \cap \mathcal{E}^c = \bigcup_{l=1}^5 \mathcal{B}_k^{(l)},$$

where $\mathcal{B}_k^{(l)}$, $l = 1, \ldots, 5$, are defined as

$$\mathcal{B}_k^{(1)} = \left\{ (i,j) \in \mathcal{E}^c \,\middle|\, -\epsilon_k \leq \left[ \boldsymbol{X}_k \right]_{ij} \leq 0, \; \left[ \nabla f(\boldsymbol{X}_k) \right]_{ij} \geq 0, \; \left[ \boldsymbol{P}_k \right]_{ij} < 0 \right\},$$

$$\mathcal{B}_k^{(2)} = \left\{ (i,j) \in \mathcal{E}^c \,\middle|\, -\epsilon_k \leq \left[ \boldsymbol{X}_k \right]_{ij} \leq 0, \; \left[ \nabla f(\boldsymbol{X}_k) \right]_{ij} \geq 0, \; \left[ \boldsymbol{P}_k \right]_{ij} \geq 0 \right\},$$

$$\mathcal{B}_k^{(3)} = \left\{ (i,j) \in \mathcal{E}^c \,\middle|\, \left[ \boldsymbol{X}_k \right]_{ij} < -\epsilon_k, \; \left[ \boldsymbol{P}_k \right]_{ij} < 0 \right\},$$

$$\mathcal{B}_k^{(4)} = \left\{ (i,j) \in \mathcal{E}^c \,\middle|\, \left[ \boldsymbol{X}_k \right]_{ij} < -\epsilon_k, \; \left[ \boldsymbol{P}_k \right]_{ij} \geq 0 \right\},$$

$$\mathcal{B}_k^{(5)} = \left\{ (i,j) \in \mathcal{E}^c \,\middle|\, \left[ \boldsymbol{X}_k \right]_{ij} > 0 \right\}.$$

Each subset $\mathcal{B}_k^{(l)}$ is disjoint with each other. $\mathcal{B}_k^{(5)}$ only contains the indexes of the diagonal elements of $\boldsymbol{X}_k$, whiles the other subsets include the indexes of the off-diagonal elements. Then one has, for any $(i,j) \in \mathcal{B}_k^{(1)}$ and $\gamma_k > 0$,

$$0 \leq \left[ \boldsymbol{X}_k(\gamma_k) \right]_{ij} - \left[ \boldsymbol{X}_k \right]_{ij} = \left[ \mathcal{P}_\Omega \left( \boldsymbol{X}_k - \gamma_k \boldsymbol{P}_k \right) \right]_{ij} - \left[ \boldsymbol{X}_k \right]_{ij} \leq -\gamma_k \left[ \boldsymbol{P}_k \right]_{ij}. \tag{30}$$

For any $(i,j) \in \mathcal{B}_k^{(2)}$ and $\gamma_k > 0$, one has $\left[ \boldsymbol{X}_k(\gamma_k) \right]_{ij} - \left[ \boldsymbol{X}_k \right]_{ij} = -\gamma_k \left[ \boldsymbol{P}_k \right]_{ij}$. For any $(i,j) \in \mathcal{B}_k^{(3)}$, if the step size satisfies

$$0 < \gamma_k \leq \min_{(i,j) \in \mathcal{B}_k^{(3)}} \frac{\epsilon_k}{\left| \left[ \boldsymbol{P}_k \right]_{ij} \right|}, \tag{31}$$

then one has $\left[ \boldsymbol{X}_k(\gamma_k) \right]_{ij} - \left[ \boldsymbol{X}_k \right]_{ij} = -\gamma_k \left[ \boldsymbol{P}_k \right]_{ij}$. Similarly, for any $(i,j) \in \mathcal{B}_k^{(4)}$ and $\gamma_k > 0$, $\left[ \boldsymbol{X}_k(\gamma_k) \right]_{ij} - \left[ \boldsymbol{X}_k \right]_{ij} = -\gamma_k \left[ \boldsymbol{P}_k \right]_{ij}$. Finally, for any $(i,j) \in \mathcal{B}_k^{(5)}$, $\left[ \boldsymbol{X}_k \right]_{ij}$ must be on the diagonal of $\boldsymbol{X}_k$. Therefore, we can directly remove the projection $\mathcal{P}_\Omega$, and obtain $\left[ \boldsymbol{X}_k(\gamma_k) \right]_{ij} - \left[ \boldsymbol{X}_k \right]_{ij} = -\gamma_k \left[ \boldsymbol{P}_k \right]_{ij}$. Therefore, if (31) holds, one has

$$\left\langle \left[ \nabla f(\boldsymbol{X}_k) \right]_{\mathcal{I}_k^c}, \; \left[ \boldsymbol{X}_k(\gamma_k) \right]_{\mathcal{I}_k^c} - \left[ \boldsymbol{X}_k \right]_{\mathcal{I}_k^c} \right\rangle \leq -\gamma_k \left\langle \left[ \nabla f(\boldsymbol{X}_k) \right]_{\mathcal{I}_k^c}, \; \left[ \boldsymbol{P}_k \right]_{\mathcal{I}_k^c} \right\rangle, \tag{32}$$

where the inequality follows from (30). Recall that $\left[ \boldsymbol{X}_k(\gamma_k) \right]_{\mathcal{I}_k} = \boldsymbol{0}$, which can be equivalently expressed as the form of projected gradient descent:

$$\left[ \boldsymbol{X}_k(\gamma_k) \right]_{\mathcal{I}_k} = \mathcal{P}_\Omega \left( \left[ \boldsymbol{X}_k \right]_{\mathcal{I}_k} - \gamma_k \left[ \tilde{\boldsymbol{D}}_k \odot \nabla f(\boldsymbol{X}_k) \right]_{\mathcal{I}_k} \right), \tag{33}$$

where we write $\mathcal{P}_\Omega \left( [\boldsymbol{A}]_{\mathcal{I}_k} \right)$ for $\left[ \mathcal{P}_\Omega(\boldsymbol{A}) \right]_{\mathcal{I}_k}$ with a slight abuse of notation, and $\tilde{\boldsymbol{D}}_k$ is defined as

$$\left[ \tilde{\boldsymbol{D}}_k \right]_{ij} = \frac{\epsilon_k}{\gamma_k \left| [\nabla f(\boldsymbol{X}_k)]_{ij} \right|}, \quad \forall (i,j) \in \mathcal{I}_k \setminus \mathcal{E}.$$

Then one has

$$\left\| \boldsymbol{X}_k(\gamma_k) - \boldsymbol{X}_k \right\|_{\mathrm{F}}^2$$
$$\leq \gamma_k^2 \left\langle \left[ \boldsymbol{P}_k \right]_{\mathcal{I}_k^c}, \; \left[ \boldsymbol{P}_k \right]_{\mathcal{I}_k^c} \right\rangle - \gamma_k \left\langle \left[ \tilde{\boldsymbol{D}}_k \odot \nabla f(\boldsymbol{X}_k) \right]_{\mathcal{I}_k}, \; \left[ \boldsymbol{X}_k(\gamma_k) \right]_{\mathcal{I}_k} - \left[ \boldsymbol{X}_k \right]_{\mathcal{I}_k} \right\rangle \tag{34}$$
$$\leq \gamma_k^2 \left\langle \left[ \boldsymbol{P}_k \right]_{\mathcal{I}_k^c}, \; \left[ \boldsymbol{P}_k \right]_{\mathcal{I}_k^c} \right\rangle + a_k \left\langle \left[ \nabla f(\boldsymbol{X}_k) \right]_{\mathcal{I}_k}, \; \left[ \boldsymbol{X}_k \right]_{\mathcal{I}_k} \right\rangle,$$

where $a_k = \dfrac{\epsilon_k}{\left\| \left[ \nabla f(\boldsymbol{X}_k) \right]_{\mathcal{T}_k \setminus \mathcal{E}} \right\|_{\min}}$, and $\mathcal{T}_k$ represents $\mathcal{T}(\boldsymbol{X}_k, \epsilon_k)$. Note that $\mathcal{I}_k = \mathcal{T}_k \cup \mathcal{E}$, and $[\boldsymbol{X}_k]_{ij} = [\boldsymbol{X}_k(\gamma_k)]_{ij} = 0$ for any $(i,j) \in \mathcal{E}$.

It is worth mentioning that we primarily focus on cases with non-empty $\mathcal{T}_k$. In the rare event that $\mathcal{T}_k$ is empty, such as when $\mathcal{T}_\delta$ is empty (in this situation we define $\epsilon_k = \delta$, leading to an empty $\mathcal{T}_k$), we can simply ignore the terms related to $\mathcal{I}_k$ in (34) and the following proof.

Furthermore, one has

$$\left\langle \left[ \boldsymbol{P}_k \right]_{\mathcal{I}_k^c}, \; \left[ \boldsymbol{P}_k \right]_{\mathcal{I}_k^c} \right\rangle \leq \left\| \left[ \boldsymbol{M}_k^{-1} \right]_{\mathcal{I}_k^c \mathcal{I}_k^c}^{\frac{1}{2}} \right\|_2^2 \left\| \left[ \boldsymbol{M}_k^{-1} \right]_{\mathcal{I}_k^c \mathcal{I}_k^c}^{\frac{1}{2}} \left[ \nabla f(\boldsymbol{X}_k) \right]_{\mathcal{I}_k^c} \right\|^2$$
$$= \lambda_{\max} \left( \left[ \boldsymbol{M}_k^{-1} \right]_{\mathcal{I}_k^c \mathcal{I}_k^c} \right) \left\langle \left[ \nabla f(\boldsymbol{X}_k) \right]_{\mathcal{I}_k^c}, \; \left[ \boldsymbol{P}_k \right]_{\mathcal{I}_k^c} \right\rangle. \tag{35}$$

The largest eigenvalue of $\left[\boldsymbol{M}_k^{-1}\right]_{\mathcal{I}_k^c \mathcal{I}_k^c}$ can be bounded by

$$\lambda_{\max}\left(\left[\boldsymbol{M}_k^{-1}\right]_{\mathcal{I}_k^c \mathcal{I}_k^c}\right) \leq \lambda_{\max}\left(\boldsymbol{H}_k^{-1}\right) = \lambda_{\max}\left(\boldsymbol{X}_k \otimes \boldsymbol{X}_k\right) \leq M^2, \tag{36}$$

where the first inequality follows from the Eigenvalue Interlacing Theorem, and the second inequality follows from Lemma B.1. Combining (34), (35) and (36) yields

$$\left\|\boldsymbol{X}_k(\gamma_k) - \boldsymbol{X}_k\right\|_{\mathrm{F}}^2 \leq \gamma_k^2 M^2 \big\langle\left[\nabla f(\boldsymbol{X}_k)\right]_{\mathcal{I}_k^c}, \left[\boldsymbol{P}_k\right]_{\mathcal{I}_k^c}\big\rangle + a_k \big\langle\left[\nabla f(\boldsymbol{X}_k)\right]_{\mathcal{I}_k}, \left[\boldsymbol{X}_k\right]_{\mathcal{I}_k}\big\rangle.$$

Since $\boldsymbol{X}_k(\gamma_k) \in L_f$, we obtain

$$f\left(\boldsymbol{X}_k(\gamma_k)\right) - f\left(\boldsymbol{X}_k\right) \leq \big\langle\nabla f(\boldsymbol{X}_k),\, \boldsymbol{X}_k(\gamma_k) - \boldsymbol{X}_k\big\rangle + \frac{1}{2m^2}\left\|\boldsymbol{X}_k(\gamma_k) - \boldsymbol{X}_k\right\|_{\mathrm{F}}^2, \tag{37}$$

where the inequality follows from

$$\lambda_{\max}\left(\nabla^2 f(\boldsymbol{X})\right) = \lambda_{\max}\left(\boldsymbol{X}^{-1} \otimes \boldsymbol{X}^{-1}\right) = \lambda_{\max}^2(\boldsymbol{X}^{-1}) \leq \frac{1}{m^2}, \quad \forall \boldsymbol{X} \in L_f,$$

where the inequality follows from Lemma B.1.

Let $\bar{\gamma}_k = \min\left(\frac{2(1-\alpha)m^2}{M^2},\, \min_{(i,j)\in\mathcal{B}_k^{(3)}} \frac{\epsilon_k}{\left|\left[\boldsymbol{P}_k\right]_{ij}\right|},\, \frac{\lambda_{\min}(\boldsymbol{Z}_k)}{\rho(\boldsymbol{G}_k)}\right)$. Note that $\bar{\gamma}_k$ is bounded away from zero, because each $\left[\boldsymbol{P}_k\right]_{ij}$ is bounded following from Lemma B.1. To this end, for any $\gamma_k \in (0, \bar{\gamma}_k)$,

$$f\left(\boldsymbol{X}_k(\gamma_k)\right) - f\left(\boldsymbol{X}_k\right) \leq -\alpha\gamma_k\big\langle\left[\nabla f(\boldsymbol{X}_k)\right]_{\mathcal{I}_k^c}, \left[\boldsymbol{P}_k\right]_{\mathcal{I}_k^c}\big\rangle - \alpha\big\langle\left[\nabla f(\boldsymbol{X}_k)\right]_{\mathcal{I}_k}, \left[\boldsymbol{X}_k\right]_{\mathcal{I}_k}\big\rangle, \tag{38}$$

where the inequality follows from (32), the definition of $\epsilon_k$ in (9), the inequality $\left\|\left[\nabla f(\boldsymbol{X}_k)\right]_{\mathcal{T}_\delta \setminus \mathcal{E}}\right\|_{\min} \leq \left\|\left[\nabla f(\boldsymbol{X}_k)\right]_{\mathcal{T}_k \setminus \mathcal{E}}\right\|_{\min}$, because $\mathcal{T}_k \subseteq \mathcal{T}_\delta$, and the inequality $\big\langle\left[\nabla f(\boldsymbol{X}_k)\right]_{\mathcal{I}_k^c}, \left[\boldsymbol{P}_k\right]_{\mathcal{I}_k^c}\big\rangle \geq 0$, because $\left[\boldsymbol{M}_k^{-1}\right]_{\mathcal{I}_k^c \mathcal{I}_k^c}$ is positive definite and

$$\big\langle\left[\nabla f(\boldsymbol{X}_k)\right]_{\mathcal{I}_k^c}, \left[\boldsymbol{P}_k\right]_{\mathcal{I}_k^c}\big\rangle = \big\langle\left[\nabla f(\boldsymbol{X}_k)\right]_{\mathcal{I}_k^c}, \left[\boldsymbol{M}_k^{-1}\right]_{\mathcal{I}_k^c \mathcal{I}_k^c}\left[\nabla f(\boldsymbol{X}_k)\right]_{\mathcal{I}_k^c}\big\rangle, \tag{39}$$

completing the proof. $\qquad\square$

## B.4 Proof of Proposition 3.5

*Proof.* The line search condition (15) leads to

$$f\left(\boldsymbol{X}_k(\gamma_k)\right) - f\left(\boldsymbol{X}_k\right) \leq -\alpha\gamma_k\big\langle\left[\nabla f(\boldsymbol{X}_k)\right]_{\mathcal{I}_k^c}, \left[\boldsymbol{M}_k^{-1}\right]_{\mathcal{I}_k^c \mathcal{I}_k^c}\left[\nabla f(\boldsymbol{X}_k)\right]_{\mathcal{I}_k^c}\big\rangle, \tag{40}$$

where the inequality follows from

$$\big\langle\left[\nabla f(\boldsymbol{X}_k)\right]_{\mathcal{I}_k}, \left[\boldsymbol{X}_k\right]_{\mathcal{I}_k}\big\rangle \geq 0.$$

We further have

$$\lambda_{\min}\left(\left[\boldsymbol{M}_k^{-1}\right]_{\mathcal{I}_k^c \mathcal{I}_k^c}\right) \geq \lambda_{\min}(\boldsymbol{H}_k^{-1}) = \lambda_{\min}(\boldsymbol{X}_k \otimes \boldsymbol{X}_k) \geq m^2, \tag{41}$$

where the first and second inequalities follow from Eigenvalue Interlacing Theorem and Lemma B.1, respectively. We complete the proof by combining (40) and (41). $\qquad\square$

## B.5 Proof of Theorem 4.1

*Proof.* We first prove that Problem (1) has at least one minimizer. It is known by the Weierstrass' extreme value theorem [50] that the set of minima is nonempty for any lower semicontinuous function with a nonempty compact lower level set. Therefore, the existence of the minimizers for Problem (1) can be guaranteed by Lemma B.1.

On the other hand, we show that Problem (1) has at most one minimizer by its strict convexity. We have $\nabla^2 f(\boldsymbol{X}) = \boldsymbol{X}^{-1} \otimes \boldsymbol{X}^{-1}$. Thus $\nabla^2 f(\boldsymbol{X}) \succ \mathbf{0}$ for any $\boldsymbol{X}$ in the feasible region of Problem

(1) defined in (16). Therefore, Problem (1) is strictly convex, and thus has at most one minimizer. Together with the existence of the minimizers, we conclude that Problem (1) has an unique minimizer.

The Lagrangian of Problem (1) is

$$\mathcal{L}(\boldsymbol{X}, \boldsymbol{Y}) = -\log \det(\boldsymbol{X}) + \mathrm{tr}(\boldsymbol{XS}) + \langle \boldsymbol{Y} - \boldsymbol{\lambda}, \boldsymbol{X} \rangle, \tag{42}$$

where $\boldsymbol{Y}$ is a KKT multiplier with $Y_{ii} = 0$ for $i \in [p]$, and $\boldsymbol{\lambda}$ is defined in (25).

For a convex optimization problem with Slater's condition holding, a pair is primal and dual optimal if and only if the KKT conditions hold. Thus, $(\boldsymbol{X}^\star, \boldsymbol{Y}^\star)$ is primal and dual optimal if and only if it satisfies the KKT conditions of (1) as below,

$$-(\boldsymbol{X}^\star)^{-1} + \boldsymbol{S} - \boldsymbol{\lambda} + \boldsymbol{Y}^\star = \boldsymbol{0}; \tag{43}$$
$$X_{ij}^\star = 0, \ \forall \, (i,j) \in \mathcal{E}; \tag{44}$$
$$X_{ij}^\star Y_{ij}^\star = 0, \ X_{ij}^\star \leq 0, \ Y_{ij}^\star \geq 0, \ \forall \, i \neq j \text{ and } (i,j) \notin \mathcal{E}; \tag{45}$$
$$Y_{ii}^\star = 0, \ \forall \, i \in [p]. \tag{46}$$

We first prove that the minimizer $\boldsymbol{X}^\star$ must satisfy all conditions in (18). Note that the KKT conditions (43)-(46) must hold for the minimizer $\boldsymbol{X}^\star$. Let $\mathcal{V} = \{(i,j) \in [p]^2 \,|\, X_{ij}^\star = 0\}$. First, $X_{ij}^\star = 0$ for any $(i,j) \in \mathcal{E}$ following from (44). Second, for any $(i,j)$ with $i \neq j$ and $(i,j) \in \mathcal{V}^c$, we have $X_{ij}^\star \neq 0$ and $(i,j) \notin \mathcal{E}$. Following from (45), we further obtain $Y_{ij}^\star = 0$. Together with (46), we conclude that $Y_{ij}^\star = 0$ for any $(i,j) \in \mathcal{V}^c$. Following from (43), $\left[\nabla f(\boldsymbol{X}^\star)\right]_{\mathcal{V}^c} = \boldsymbol{0}$. Since $\boldsymbol{X}^\star$ is positive definite, $(i,i) \in \mathcal{V}^c$ for any $i \in [p]$. Then, for any $(i,j) \in \mathcal{V} \backslash \mathcal{E}$, we have $i \neq j$, thus obtain $Y_{ij}^\star \geq 0$ according to (45). Following from (43), we get $\left[\nabla f(\boldsymbol{X}^\star)\right]_{\mathcal{V} \backslash \mathcal{E}} \leq \boldsymbol{0}$.

Now we prove that any point $\boldsymbol{X}^\star \in \mathcal{M}^p$ satisfying the conditions in (18) must be the minimizer, *i.e.*, the KKT conditions (43)-(46) hold for $\boldsymbol{X}^\star$. We construct $\boldsymbol{Y}^\star$ by $\boldsymbol{Y}^\star = -\nabla f(\boldsymbol{X}^\star)$. First, it is straightforward to check that the conditions (43) and (44) hold. Second, we have

$$[\boldsymbol{Y}^\star]_{\mathcal{V}^c} = -[\nabla f(\boldsymbol{X}^\star)]_{\mathcal{V}^c} = \boldsymbol{0}. \tag{47}$$

We know that $(i,i) \in \mathcal{V}^c$ for any $i \in [p]$, since $\boldsymbol{X}^\star \in \mathcal{M}^p$. Together with (47), we obtain that the condition (46) holds.

Finally, following from the fact that $(i,i) \in \mathcal{V}^c$ for any $i \in [p]$ and $\mathcal{E} \subseteq \mathcal{V}$, we obtain

$$\{(i,j) \in [p]^2 \,|\, i \neq j, \ (i,j) \notin \mathcal{E}\} = \mathcal{I}_1 \cup \mathcal{I}_2,$$

where $\mathcal{I}_1 = \{(i,j) \in [p]^2 \,|\, (i,j) \in \mathcal{V}^c, i \neq j\}$, and $\mathcal{I}_2 = \{(i,j) \in [p]^2 \,|\, (i,j) \in \mathcal{V}, (i,j) \notin \mathcal{E}\}$. For any $(i,j) \in \mathcal{I}_1$, we have $Y_{ij}^\star = 0$ according to (47), and $X_{ij}^\star < 0$ since $\boldsymbol{X}^\star \in \mathcal{M}^p$. Thus the condition (45) holds for any $(i,j) \in \mathcal{I}_1$. For any $(i,j) \in \mathcal{I}_2$, we have $X_{ij}^\star = 0$, and $Y_{ij}^\star = -[\nabla f(\boldsymbol{X}^\star)]_{ij} \geq 0$ according to the second condition in (18). Thus the condition (45) also holds for any $(i,j) \in \mathcal{I}_2$. Totally, the condition (45) holds. To sum up, all KKT conditions (43)-(46) hold for any $\boldsymbol{X}^\star \in \mathcal{M}^p$ satisfying the conditions in (18), and thus we conclude that $\boldsymbol{X}^\star$ is the minimizer of Problem (1). $\square$

### B.6 Proof of Theorem 4.2

*Proof.* Following from Proposition 3.4, for any iterate $\boldsymbol{X}_k \in L_f$, a small enough step size ensures that the line search condition (15) holds, which leads to a sufficient decrease of the objective function as shown in Theorem 3.5, and the next iterate $\boldsymbol{X}_{k+1} \in L_f$. When the sequence starts with $\boldsymbol{X}_0 \in L_f$, each point of the sequence $\{\boldsymbol{X}_k\}_{k \geq 0}$ admits $\boldsymbol{X}_k \in L_f$. Note that it is easy to construct an initial point $\boldsymbol{X}_0 \in L_f$, because we could consider an $\boldsymbol{X}^o \in \mathcal{U}^p$ in (17), which is close to $\mathrm{cl}(\mathcal{U}^p) \setminus \mathcal{U}^p$ such that $f(\boldsymbol{X}^o)$ sufficiently large, following from the fact that $f(\boldsymbol{X}) = +\infty$ for any $\boldsymbol{X} \in \mathrm{cl}(\mathcal{U}^p) \setminus \mathcal{U}^p$. Lemma B.1 shows that the lower level set $L_f$ is compact, thus the sequence $\{\boldsymbol{X}_k\}$ has at least one limit point. For every limit $\boldsymbol{X}^\star$ of the sequence $\{\boldsymbol{X}_k\}$, we have $\boldsymbol{X}^\star \in \mathcal{M}^p$. Define $\mathcal{I}^\star = \mathcal{T}(\boldsymbol{X}^\star, \epsilon^\star) \cup \mathcal{E}$, where $\mathcal{T}(\boldsymbol{X}^\star, \epsilon^\star)$ is equal to

$$\mathcal{T}(\boldsymbol{X}^\star, \epsilon^\star) = \{(i,j) \in [p]^2 \,|\, -\epsilon^\star \leq \left[\boldsymbol{X}^\star\right]_{ij} \leq 0, \ \left[\nabla f(\boldsymbol{X}^\star)\right]_{ij} < 0\},$$

and $\epsilon^\star$ is defined as $\epsilon^\star := \min\left(2(1-\alpha)m^2\big\|\big[\nabla f(\boldsymbol{X}^\star)\big]_{\mathcal{T}_\delta^\star \backslash \mathcal{E}}\big\|_{\min},\ \delta\right)$, where $\mathcal{T}_\delta^\star$ denotes $\mathcal{T}(\boldsymbol{X}^\star, \delta)$. Since $f$ is continuous and $\{f(\boldsymbol{X}_k)\}$ keeps decreasing and is bounded, it follows that $\{f(\boldsymbol{X}_k)\}$ converges and

$$\lim_{k\to +\infty} f(\boldsymbol{X}_k) - f(\boldsymbol{X}_{k+1}) = 0.$$

Following from Proposition 3.4, the line search condition (15) ensures that each $\boldsymbol{X}_k \in \mathbb{S}_{++}^p$ and, for a strictly positive $\gamma_k$, it holds

$$f(\boldsymbol{X}_k) - f(\boldsymbol{X}_{k+1}) \geq \alpha\gamma_k\big\langle [\nabla f(\boldsymbol{X}_k)]_{\mathcal{I}_k^c},\ [\boldsymbol{P}_k]_{\mathcal{I}_k^c}\big\rangle + \alpha\big\langle [\nabla f(\boldsymbol{X}_k)]_{\mathcal{I}_k},\ [\boldsymbol{X}_k]_{\mathcal{I}_k}\big\rangle \qquad (48)$$
$$= \alpha\gamma_k\big\langle [\nabla f(\boldsymbol{X}_k)]_{\mathcal{I}_k^c},\ \big[\boldsymbol{M}_k^{-1}\big]_{\mathcal{I}_k^c\mathcal{I}_k^c}[\nabla f(\boldsymbol{X}_k)]_{\mathcal{I}_k^c}\big\rangle + \alpha\big\langle [\nabla f(\boldsymbol{X}_k)]_{\mathcal{I}_k},\ [\boldsymbol{X}_k]_{\mathcal{I}_k}\big\rangle.$$

Recall that $\mathcal{I}_k = \mathcal{T}_k \cup \mathcal{E}$. We note that the following facts hold: $\alpha \in (0,1)$ is a constant, $\big[\boldsymbol{M}_k^{-1}\big]_{\mathcal{I}_k^c\mathcal{I}_k^c}$ is positive definite, $[\boldsymbol{X}_k]_{ij} = 0$ for any $(i,j) \in \mathcal{E}$, $[\boldsymbol{X}_k]_{ij} \leq 0$ for any $(i,j) \in \mathcal{I}_k\backslash\mathcal{E}$, and $[\nabla f(\boldsymbol{X}_k)]_{ij} < 0$ for any $(i,j) \in \mathcal{I}_k\backslash\mathcal{E}$ due to the definition of $\mathcal{T}(\boldsymbol{X}, \epsilon)$. Thus, the two terms on the right-hand side of (48) are both nonnegative. Moreover, the right-hand side of (48) approaches zero if and only if both $\big[\nabla f(\boldsymbol{X}_k)\big]_{\mathcal{I}_k^c}$ and $[\boldsymbol{X}_k]_{\mathcal{I}_k}$ simultaneously go to zero. Therefore, we deduce that every limit point $\boldsymbol{X}^\star$ must satisfy:

$$\big[\nabla f\left(\boldsymbol{X}^\star\right)\big]_{\{\mathcal{I}^\star\}^c} = \boldsymbol{0}, \quad \text{and} \quad \big[\boldsymbol{X}^\star\big]_{\mathcal{I}^\star} = \boldsymbol{0}. \qquad (49)$$

We show that every limit point $\boldsymbol{X}^\star$ is the minimizer of Problem (1) according to Theorem 4.1. Let $\mathcal{V} = \big\{(i,j) \in [p]^2 \,\big|\, \big[\boldsymbol{X}^\star\big]_{ij} = 0\big\}$. First, for any $(i,j) \in \mathcal{E}$, we have $\big[\boldsymbol{X}^\star\big]_{ij} = 0$, because of the projection $\mathcal{P}_\Omega$ in each iteration. Note that $\big[\boldsymbol{X}^\star\big]_{ij} = 0$ for any $(i,j) \in \mathcal{I}^\star$ according to (49). For any $(i,j) \in \mathcal{V}^c$, i.e., $\big[\boldsymbol{X}^\star\big]_{ij} \neq 0$, we must have $(i,j) \in \{\mathcal{I}^\star\}^c$. Together with (49), $\big[\nabla f\left(\boldsymbol{X}^\star\right)\big]_{\mathcal{V}^c} = \boldsymbol{0}$ holds. For any $(i,j) \in \mathcal{V}\backslash\mathcal{E}$, we must have

$$(i,j) \in \mathcal{T}(\boldsymbol{X}^\star, \epsilon^\star) \cup \{\mathcal{I}^\star\}^c.$$

Recall that $\big[\nabla f\left(\boldsymbol{X}^\star\right)\big]_{ij} < 0$ for any $(i,j) \in \mathcal{T}(\boldsymbol{X}^\star, \epsilon^\star)$, and $\big[\nabla f\left(\boldsymbol{X}^\star\right)\big]_{ij} = 0$ for any $(i,j) \in \{\mathcal{I}^\star\}^c$. Overall, we obtain

$$\big[\nabla f\left(\boldsymbol{X}^\star\right)\big]_{\mathcal{V}\backslash\mathcal{E}} \leq \boldsymbol{0}.$$

To sum up, all the conditions in Theorem 4.1 hold for every limit point $\boldsymbol{X}^\star$, and thus every limit point is the minimizer of Problem (1).

Since the minimizer of Problem (1) is unique, we obtain that the limit point of the sequence $\{\boldsymbol{X}_k\}$ is also unique, and thus $\{\boldsymbol{X}_k\}$ is convergent. Therefore, we conclude that the sequence $\{\boldsymbol{X}_k\}$ converges to the unique minimizer of Problem (1). The monotone decreasing of the sequence $\{f(\boldsymbol{X}_k)\}$ can be established by Proposition 3.5. $\qquad\square$

## B.7 Proof of Theorem 4.4

*Proof.* We prove that the support of $\boldsymbol{X}^\star$ is consistent with the set $\mathcal{I}_k^c$ for a sufficiently large $k$. Without loss of generality, we specify the constant $\delta$ in (9) as

$$\delta = \omega \min_{(i,j)\in\text{supp}(\boldsymbol{X}^\star)} \big|\big[\boldsymbol{X}^\star\big]_{ij}\big|, \qquad (50)$$

where $\omega \in (0,1)$ is a constant. Note that the sequence $\{\boldsymbol{X}_k\}$ converges to $\boldsymbol{X}^\star$. Under Theorem 4.3 that $\big[\nabla f(\boldsymbol{X}^\star)\big]_{ij}$ is strictly negative for any $(i,j) \in \text{supp}^c(\boldsymbol{X}^\star) \setminus \mathcal{E}$, there must exist some $a > 0$ and $K_1 \in \mathbb{N}_+$ such that

$$\big[\nabla f(\boldsymbol{X}_k)\big]_{ij} < -\frac{a}{2(1-\alpha)m^2}, \quad \forall\,(i,j) \in \text{supp}^c(\boldsymbol{X}^\star) \setminus \mathcal{E} \qquad (51)$$

holds for any $k \geq K_1$, where $\alpha \in (0,1)$ is a constant, and $m$ is a positive constant defined in Lemma B.1. We consider a neighbourhood of $\boldsymbol{X}^\star$ defined by

$$\mathcal{N}\big(\boldsymbol{X}^\star; r\big) := \big\{\boldsymbol{X} \in \mathbb{R}^{p\times p} \,\big|\, \big\|\boldsymbol{X} - \boldsymbol{X}^\star\big\|_{\mathrm{F}} \leq r\big\}, \qquad (52)$$

where $r$ is a positive constant defined as

$$r := \min\Big(c \min_{(i,j)\in\mathrm{supp}(\boldsymbol{X}^\star)}\big|\big[\boldsymbol{X}^\star\big]_{ij}\big|,\, a,\, \delta\Big), \tag{53}$$

where $c < 1 - \omega$ is a positive constant. There must exist $K_2 \in \mathbb{N}_+$ such that $\boldsymbol{X}_k \in \mathcal{N}\big(\boldsymbol{X}^\star; r\big)$ holds for any $k \geq K_2$. Take $K_o = \max(K_1, K_2)$. For any $k \geq K_o$ and $(i,j) \in \mathrm{supp}(\boldsymbol{X}^\star)$, one has

$$\big|\big[\boldsymbol{X}^\star\big]_{ij}\big| - \big|\big[\boldsymbol{X}_k\big]_{ij}\big| \leq \big|\big[\boldsymbol{X}^\star\big]_{ij} - \big[\boldsymbol{X}_k\big]_{ij}\big| \leq \big\|\boldsymbol{X}_k - \boldsymbol{X}^\star\big\|_{\mathrm{F}} \leq r,$$

Thus one can obtain, for any $k \geq K_o$,

$$\big|\big[\boldsymbol{X}_k\big]_{ij}\big| \geq \min_{(i,j)\in\mathrm{supp}(\boldsymbol{X}^\star)}\big|\big[\boldsymbol{X}^\star\big]_{ij}\big| - r > \delta > 0, \quad \forall\,(i,j) \in \mathrm{supp}(\boldsymbol{X}^\star), \tag{54}$$

where the last inequality follows from (50) and (53). Recall that for any $(i,j) \in \mathcal{T}(\boldsymbol{X}_k, \delta)$, $\big|\big[\boldsymbol{X}_k\big]_{ij}\big| \leq \delta$. Then one has

$$\mathcal{T}(\boldsymbol{X}_k, \delta) \subseteq \mathrm{supp}^c(\boldsymbol{X}^\star). \tag{55}$$

Then the $\epsilon_k$ in (9) can be bounded by

$$\epsilon_k \geq \min\Big(2(1-\alpha)m^2\big\|\big[\nabla f(\boldsymbol{X}_k)\big]_{\mathrm{supp}^c(\boldsymbol{X}^\star)\setminus\mathcal{E}}\big\|_{\min},\, \delta\Big) \geq \min(a,\, \delta), \tag{56}$$

where the first and second inequalities follow from (55) and (51), respectively. For any $k \geq K_o$ and $(i,j) \in \mathcal{T}(\boldsymbol{X}_k, \epsilon_k) \cup \mathcal{E}$, one has

$$\big|\big[\boldsymbol{X}_k\big]_{ij}\big| \leq \epsilon_k \leq \delta,$$

where the second inequality follows from the definition of $\epsilon_k$. Thus we obtain

$$\mathcal{T}(\boldsymbol{X}_k, \epsilon_k) \cup \mathcal{E} \subseteq \mathrm{supp}^c(\boldsymbol{X}^\star). \tag{57}$$

On the other hand, for any $k \geq K_o$ and $(i,j) \in \mathrm{supp}^c(\boldsymbol{X}^\star)$, one has

$$\big|\big[\boldsymbol{X}_k\big]_{ij}\big| = \big|\big[\boldsymbol{X}_k\big]_{ij} - \big[\boldsymbol{X}^\star\big]_{ij}\big| \leq \big\|\boldsymbol{X}_k - \boldsymbol{X}^\star\big\|_{\mathrm{F}} \leq r \leq \epsilon_k, \tag{58}$$

where the last inequality follows from (53) and (56). Therefore, one has

$$\mathrm{supp}^c(\boldsymbol{X}^\star) \subseteq \mathcal{T}(\boldsymbol{X}_k, \epsilon_k) \cup \mathcal{E} \cup \mathcal{B}_k^{(1)} \cup \mathcal{B}_k^{(2)}.$$

Note that $\mathcal{B}_k^{(5)} \cap \mathrm{supp}^c(\boldsymbol{X}^\star) = \varnothing$, because any $(i,j) \in \mathcal{B}_k^{(5)}$ corresponds to the element on the diagonal which must be nonzero. Moreover, following from (51), one has

$$\big(\mathrm{supp}^c(\boldsymbol{X}^\star) \setminus \mathcal{E}\big) \cap \mathcal{B}_k^{(1)} = \varnothing, \quad \text{and} \quad \big(\mathrm{supp}^c(\boldsymbol{X}^\star) \setminus \mathcal{E}\big) \cap \mathcal{B}_k^{(2)} = \varnothing.$$

Therefore, we can obtain $\mathrm{supp}^c(\boldsymbol{X}^\star) \subseteq \mathcal{T}(\boldsymbol{X}_k, \epsilon_k) \cup \mathcal{E}$. Together with (57), we obtain

$$\mathrm{supp}^c(\boldsymbol{X}^\star) = \mathcal{T}(\boldsymbol{X}_k, \epsilon_k) \cup \mathcal{E} = \mathcal{I}_k. \tag{59}$$

Equivalently, we have

$$\mathrm{supp}(\boldsymbol{X}^\star) = \mathcal{I}_k^c, \quad \forall\, k \geq K_o. \tag{60}$$

We can see that the sets $\mathcal{I}_k$ and $\mathcal{I}_k^c$ are fixed for any $k \geq K_o$. Therefore, following from the iterate of $\boldsymbol{X}_{k+1}$,

$$\big[\boldsymbol{X}_{k+1}\big]_{\mathcal{I}_{k+1}} = \big[\boldsymbol{X}_{k+1}\big]_{\mathcal{I}_k} = \boldsymbol{0}, \quad \forall\, k \geq K_o.$$

Moreover, together with (54) and (60), we obtain that for any $k \geq K_o$, $\big[\boldsymbol{X}_{k+1}\big]_{\mathcal{I}_{k+1}^c} \neq \boldsymbol{0}$. Take $k_o = K_o + 1$. We obtain

$$\mathrm{supp}(\boldsymbol{X}_k) = \mathcal{I}_k^c, \quad \forall\, k \geq k_o,$$

completing the proof. $\qquad\square$

## B.8 Proof of Theorem 4.5

*Proof.* Theorem 4.5 is a direct extension of the result in [51]. Let $g$ be twice continuously differentiable and consider the following iterate

$$\boldsymbol{x}_{x+1} = \boldsymbol{x}_k - \gamma_k \boldsymbol{D}_k \nabla g(\boldsymbol{x}_k),$$

where $\boldsymbol{D}_k$ is positive definite and symmetric. Then the sequence $\{\boldsymbol{x}_k\}$ admits the following convergence result [51],

$$\limsup_{k\to\infty} \frac{\left\|\boldsymbol{x}_{k+1} - \boldsymbol{x}^\star\right\|^2_{\boldsymbol{D}_k^{-1}}}{\left\|\boldsymbol{x}_k - \boldsymbol{x}^\star\right\|^2_{\boldsymbol{D}_k^{-1}}} = \limsup_{k\to\infty} \max\left(|1 - \gamma_k m'_k|^2, |1 - \gamma_k M'_k|^2\right), \tag{61}$$

where $\boldsymbol{x}^\star$ is the limit of the sequence $\{\boldsymbol{x}_k\}$, which satisfies that $\nabla g(\boldsymbol{x}^\star) = \boldsymbol{0}$ and $\nabla^2 g(\boldsymbol{x}^\star)$ is positive definite, and $m'_k$ and $M'_k$ are the smallest and largest eigenvalues of $(\boldsymbol{D}_k)^{1/2} \nabla^2 g(\boldsymbol{x}_k)(\boldsymbol{D}_k)^{1/2}$, respectively. This conclusion is extended from the convergence result for the quadratic objective function. Conceptually, this makes sense because a twice continuously differentiable objective function is very close to a positive definite quadratic function in the neighborhood of a non-singular local minimum.

Following from Theorem 4.4, for any $k \geq k_o$, iterate (13) can be written as

$$\left[\boldsymbol{X}_{k+1}\right]_{\mathcal{I}_k^c} = \left[\boldsymbol{X}_k\right]_{\mathcal{I}_k^c} - \gamma_k \boldsymbol{R}_k^{-1} \left[\nabla f(\boldsymbol{X}_k)\right]_{\mathcal{I}_k^c}, \tag{62}$$

which reduces to an iterate of an unconstrained optimization algorithm on some subspace. Following from the results in (61), we obtain

$$\limsup_{k\to\infty} \frac{\left\|\left[\boldsymbol{X}_{k+1}\right]_{\mathcal{I}_k^c} - \left[\boldsymbol{X}^\star\right]_{\mathcal{I}_k^c}\right\|^2_{\boldsymbol{R}_k}}{\left\|\left[\boldsymbol{X}_k\right]_{\mathcal{I}_k^c} - \left[\boldsymbol{X}^\star\right]_{\mathcal{I}_k^c}\right\|^2_{\boldsymbol{R}_k}} = \limsup_{k\to\infty} \max\left(|1 - \gamma_k m_k|^2, |1 - \gamma_k M_k|^2\right), \tag{63}$$

where $m_k$ and $M_k$ are the smallest and largest eigenvalues of $\boldsymbol{R}_k^{-\frac{1}{2}} \left[\boldsymbol{H}_k\right]_{\mathcal{I}_k^c \mathcal{I}_k^c} \boldsymbol{R}_k^{-\frac{1}{2}}$, respectively. Following from (11), Theorem 4.2, and $\boldsymbol{R}_k^{-1} = \left[\boldsymbol{H}_k^{-1}\right]_{\mathcal{I}_k^c \mathcal{I}_k^c}$, we obtain

$$\left\|\left[\boldsymbol{X}_k\right]_{\mathcal{I}_k^c} - \left[\boldsymbol{X}^\star\right]_{\mathcal{I}_k^c}\right\|^2_{\boldsymbol{R}_k} = \left\|\boldsymbol{X}_k - \boldsymbol{X}^\star\right\|^2_{\boldsymbol{M}_k}, \quad \forall k \geq k_o. \tag{64}$$

When $k \geq k_o$, the line search condition reduces to

$$f\left(\boldsymbol{X}_{k+1}\right) \leq f\left(\boldsymbol{X}_k\right) - \alpha \gamma_k \left\langle \left[\nabla f(\boldsymbol{X}_k)\right]_{\mathcal{I}_k^c}, \left[\boldsymbol{P}_k\right]_{\mathcal{I}_k^c}\right\rangle.$$

Similar to the unconstrained case in [52], the step size must satisfy the line search condition if $\gamma_k \geq \min\left(1, 2(1-\alpha)\beta/M_k\right)$ as $k \to \infty$. Then together with (63) and (64), we complete the proof. $\square$