# OpenReview forum: "Fast Projected Newton-like Method for Precision Matrix Estimation under Total Positivity"
_NeurIPS.cc/2023/Conference — NeurIPS 2023 poster_

### Official Review · Reviewer_oo93 · 2023-06-20

**Soundness:** 3 good
**Presentation:** 3 good
**Contribution:** 3 good
**Rating:** 6
**Confidence:** 3

**Summary:**

This paper proposed a novel algorithm for solving the problem of estimating precision matrices in Gaussian distributions with MTP2 constraint. The problem formulation the paper considered is a sign-constrained log-determinant program with disconnectivity constraints. The algorithm is based on a carefully designed search direction and variable partitioning scheme. The author provide the choices of approximate Newton direction and the computing step size with the theoretical convergence analysis, where the author finally shows the convergence in finite iterations. The experimental results indicates better computational efficiency of the proposed algorithm compared to the state-of-the-art methods.

**Strengths:**

The paper is well written and clear, the approach is well supported by the theoretical analysis. The experiment and application seem strong and nice. The authors also made comparisons to other state-of-the-art methods that makes the paper more complete.

**Weaknesses:**

- Some quantities could be elaborated in the main paper. e.g., $D_k$ in line 153. I am not sure if the form of the diagonal matrix is important or a minor issue that will only be useful for theoretical analysis.

- It might also be good to mention the special case of the algorithm without the disconnectivity constraint and how it compares with other existing methods.

- It seems to me that the choice of regularization parameter $\lambda_{ij}$ doesn't affect the results in Theorem 4.4 & 4.5. Could you elaborate more on this?



**Questions:**

See weakness.

**Limitations:**

The authors have adequately addressed the limitations.

---

> ### Author Rebuttal · Authors · 2023-08-08
>
> > Strengths:
> > The paper is well written and clear, the approach is well supported by the theoretical analysis. The experiment and application seem strong and nice. The authors also made comparisons to other state-of-the-art methods that makes the paper more complete.
>
> **Reply:** Thank you for your kind words and for acknowledging the clarity, theoretical support, experimental strength, and comparisons made in our paper. We are grateful for your time and efforts in reviewing our paper and providing us with your valuable feedback.
>
> > Weaknesses:
> > 1. Some quantities could be elaborated in the main paper. e.g., $D_k$ in line 153. I am not sure if the form of the diagonal matrix is important or a minor issue that will only be useful for theoretical analysis.
>
> **Reply:** Thank you for your insightful comment.  The primary purpose of constructing this diagonal matrix is to ensure the theoretical convergence of our algorithm. In practical implementation, however, there is no need to explicitly compute this diagonal matrix. This crucial distinction between the theoretical role and practical implementation of the diagonal matrix is indeed worth emphasizing.
>
> We appreciate your suggestion and will clarify this point in our revised manuscript.
>
> > 2. It might also be good to mention the special case of the algorithm without the disconnectivity constraint and how it compares with other existing methods.
>
> **Reply:** We appreciate your suggestion to emphasize the performance of our algorithm when the disconnectivity constraint is not applied. Indeed, such experimental comparisons have been conducted and are displayed in Figures 2 and 3, demonstrating the superior computational efficiency of our algorithm relative to existing methods.
>
> In light of your valuable feedback, we will refine our manuscript to better highlight this aspect, emphasizing the robustness and efficiency of our algorithm under various conditions.
>
> > 3. It seems to me that the choice of regularization parameter $\lambda_{ij}$ doesn't affect the results in Theorem 4.4 \& 4.5. Could you elaborate more on this?
>
> **Reply:** Thank you for your insightful observation regarding the role of the regularization parameter in Theorems 4.4 and 4.5.
>
> You're correct in noting that the choice of the regularization parameter does not impact the conclusion of these theorems. This is primarily because it acts as a linear term in our optimization problem. Consequently, it does not influence the convergence behavior of our algorithm.
>
> We appreciate your constructive comment. In our revised manuscript, we will provide a more detailed explanation on this matter.

---

> > ### Comment · Reviewer_oo93 · 2023-08-16
> >
> > I would like to thank the authors for their reply. My assessment remains inclined to the positive.

---

> > > ### Author Response · Authors · 2023-08-18
> > >
> > > We greatly appreciate your positive assessment and your time in reviewing our manuscript. Your feedback has been invaluable in improving our work.

---

### Official Review · Reviewer_KQq4 · 2023-07-03

**Soundness:** 4 excellent
**Presentation:** 3 good
**Contribution:** 4 excellent
**Rating:** 7
**Confidence:** 3

**Summary:**

This paper presents a novel projected Newton-like method for computing the inverse covariance matrix of a multivariate Gaussian distribution, for which the off-diagonal elements are non-positive. The method is shown to be an order of magnitude faster than existing state-of-the-art methods for this task, while the authors also prove convergence to the global optimum.

**Strengths:**

Well-written.
Fast algorithm.
Complete theory.
Convincing experiments.

**Weaknesses:**

I would like the authors to spend a little more time motivating and introducing the problem.
In particular, discussing the following questions on the level of introduction, might make the paper more appealing.
1. What features of the data lead to such an inverse covariance matrix structure?
2. What is the exact "reduction on the number of observations" for a model with the MTP_2 structure? (line 23).
3. What is the role, in terms of data / applications, of the disconnectivity set Epsilon?
4. Why is problem (1) preferred to other alternatives?


**Questions:**

Please see above.

**Limitations:**

-

---

> ### Author Rebuttal · Authors · 2023-08-08
>
> > I would like the authors to spend a little more time motivating and introducing the problem. In particular, discussing the following questions on the level of introduction, might make the paper more appealing.
>
> **Reply:** Thank you for your valuable feedback and constructive suggestions. We appreciate your recommendation to provide a more detailed introduction to motivate the problem and provide context. Accordingly, we commit to expanding our introduction in the revised manuscript to address each of your suggested questions, ultimately making the paper more appealing and comprehensive.
>
> > 1. What features of the data lead to such an inverse covariance matrix structure?
>
> **Reply:** Thank you for your insightful query. The structure of the inverse covariance matrix is primarily determined by the inherent relationships within the data. In particular, conditional independence among variables is represented as zeroes in this matrix. This structure can be seen in data with inherent sparsity, where only a small subset of variables directly influences a given variable. In the revised manuscript, we will elaborate on how the nature of the data leads to such a structure.
>
> We will provide a more detailed explanation of this in the revised manuscript.
>
> > 2. What is the exact "reduction on the number of observations" for a model with the $\mathrm{MTP}_2$ structure? (line 23).
>
> **Reply:** Thank you for your pertinent query. In the context of a Gaussian distribution, the $\mathrm{MTP}_2$ structure leads to a significant reduction in the number of observations required for the existence of a maximum likelihood estimator (MLE). Specifically, the requirement reduces from $n \geq p$ to $n \geq 2$ [R1, R2], where $n$ and $p$ represent the number of observations and the problem dimension, respectively.
>
> In the case of Ising models, the MLE can exist with only $n = p$ observations under $\mathrm{MTP}_2$ constraints [R3], which is a substantial reduction from the $2^p$ observations required without these constraints.
>
> We will provide a more detailed explanation of this aspect in the revised manuscript.
>
> References:
>
> [R1] M. Slawski, and M. Hein. "Estimation of positive definite M-matrices and structure learning for attractive Gaussian Markov random fields." Linear Algebra and its Applications, 473: 145-179, 2015.
>
> [R2] S. Lauritzen, C. Uhler, and P. Zwiernik. "Maximum likelihood estimation in Gaussian models under total positivity." Annals of Statistics, 47(4): 1835-1863, 2019.
>
> [R3] S. Lauritzen, C. Uhler, and P. Zwiernik. "Total positivity in exponential families with application to binary variables." Annals of Statistics, 49(3): 1436-1459, 2021.
>
> > 3. What is the role, in terms of data / applications, of the disconnectivity set Epsilon?
>
> **Reply:** Thank you for your insightful question. The disconnectivity set plays a crucial role in integrating prior insights about the structure of the model. It enables us to incorporate pre-existing knowledge about inherent disconnections in the data, especially in cases where certain variables are known to be conditionally independent.
>
> Furthermore, it is particularly significant in learning structured graphs, such as bipartite graphs, where nodes are partitioned into two disjoint sets, leading to inherent disconnection within each set.
>
> We will include expanded discussions on the role of the disconnectivity set in our revised manuscript.
>
> > 4. Why is problem (1) preferred to other alternatives?
>
> **Reply:** Thank you for your pertinent question. Problem (1) is preferred due to its unique characteristics and benefits. It is designed to minimize the Gaussian negative log-likelihood, regularized by the weighted $\ell_1$-norm of the off-diagonal entries of the precision matrix. As a result, we obtain a log-determinant program that can be solved in polynomial time using various methods. This estimator has been well-studied regarding its behavior in high-dimensional scenarios, particularly when dealing with multivariate Gaussian data. In this context, the non-zero structure of the precision matrix encapsulates conditional dependence. Interestingly, this estimator remains robust even for non-Gaussian data, as it corresponds to minimizing a regularized log-determinant Bregman divergence.
>
> Moreover, Problem (1) stands out due to its ability to incorporate the disconnectivity constraint and sign constraint. The disconnectivity constraint provides a mechanism for integrating prior knowledge about the graph structure and facilitates learning structured graphs. On the other hand, the sign constraint contributes to reducing the sample size necessary for estimation, hence enhancing the effectiveness of the method.
>
> In the revised manuscript, we will delve into these aspects further to enhance the clarity and readability of our work. We appreciate your insightful question, which will undoubtedly improve the overall quality of our paper.

---

> > ### Comment · Reviewer_KQq4 · 2023-08-12
> >
> > Thank you for the detailed responses.

---

> > > ### Author Response · Authors · 2023-08-12
> > >
> > > We appreciate your acknowledgement and are glad to know that our responses were helpful. Please do not hesitate to reach out if you have more questions.

---

### Official Review · Reviewer_7MEo · 2023-07-06

**Soundness:** 2 fair
**Presentation:** 2 fair
**Contribution:** 2 fair
**Rating:** 3
**Confidence:** 4

**Summary:**

(1) This paper investigates the problem of estimating precision matrices in Gaussian distributions  that are multivariate totally positive of order two (MTP2), and propose an algorithm based on the two-metric projection method.
(2) Empirical tests are conducted.

**Strengths:**

- **Originality:** The techniques used in this paper reduces computational complexity and its theoretical convergence is established.

-  **Quality:** The notations, problem statements, and mathematical proofs appear rigorous, although I have not scrutinized them word by word.

- **Clarity:** The readability of this article is good.

- **Significance:** The techniques (and tricks) may inspire and guide future researchers.

**Weaknesses:**

1. The main issue with this paper is that it does not cite the highly relevant paper [A]. The bound provided by theoretical analysis of this paper is a supremum of a limit point, which generally can not guarantee the convergence, while the theoretical result of [A] is much more attractive.

2. Also, in the empirical study, the algorithm presented in the paper has not been compared to [A].

[A] Ying, Jiaxi, José Vinícius De Miranda Cardoso, and Daniel P. Palomar. "Adaptive Estimation of Graphical Models under Total Positivity." (2023).

**Questions:**

Please address these questions I posed to in the previous message.

**Limitations:**

Yes, they have.

---

> ### Author Rebuttal · Authors · 2023-08-09
>
> > 1a. The main issue with this paper is that it does not cite the highly relevant paper [A].
>
> > [A] J. Ying, J. V. D. M. Cardoso, and D. P. Palomar. "Adaptive Estimation of Graphical Models under Total Positivity." ICML, 2023.
>
> **Reply:** We sincerely appreciate your insightful feedback. We apologize for our initial oversight in not citing this relevant work—this was definitely not our intention to overlook its contributions to the field.
>
> **It's worth noting that paper [A] appeared after an earlier version of our work had been posted on arXiv and it cited this earlier version.**
>
> We would like to clarify that our work and paper [A] have different emphases and distinct methodologies, although both aim to advance the field of MTP2 graphical models.
>
> **Paper [A] primarily focuses on the enhancement of precision matrix estimation and graph edge recovery**, prioritizing these aspects over computational efficiency. The authors present a multiple-stage estimation method that progressively refines the estimate at each stage by solving a constrained log-determinant program, using a projected gradient descent (PGD) algorithm. However, as a first-order algorithm, PGD is often hindered by computational inefficiency due to its slow convergence rate.
>
> Conversely, **our work emphasizes enhancing computational efficiency, recognizing this as a crucial aspect when dealing with large-scale problems.** Our paper is dedicated to the development of an efficient and scalable algorithm tailored for estimating large-scale MTP2 graphical models. In comparison with the PGD algorithm used in paper [A], our proposed second-order algorithm achieves a significantly faster convergence rate, while maintaining the same per-iteration computational complexity. Our extensive experiments demonstrate that our algorithm significantly boosts computational efficiency compared to state-of-the-art methods, including PGD.
>
> **We provide a more detailed comparison of the two papers in our “global” response.** In the revised manuscript, we will ensure to appropriately cite paper [A] and clearly articulate its relevance to our work. We appreciate your constructive feedback, which helps enhance the clarity and thoroughness of our paper.
>
> > 1b. The bound provided by theoretical analysis of this paper is a supremum of a limit point, which generally can not guarantee the convergence, while the theoretical result of [A] is much more attractive.
>
> **Reply:** Thank you for your insightful feedback. In terms of the theoretical results of the two papers, we would like to provide the following clarifications. **Our theoretical analysis shows that the proposed algorithm is guaranteed to converge to the global minimizer**, with the set of free variables converging to the support of the global minimizer within finite iterations, as declared in Theorem 4.2 and Theorem 4.4, respectively. We also establish the sequence convergence rate in Theorem 4.5.
>
> In contrast, paper [A] emphasizes enhancing estimation performance. Their Theorem 4.4 characterizes the estimation error between their proposed estimator and the underlying precision matrix. However, **they fail to provide a guarantee for the algorithm's convergence to a global minimizer and do not establish the sequence convergence rate.** In fact, the formulation in paper [A] is nonconvex, and their algorithm only guarantees convergence to a stationary point. We will include a more detailed comparison in the revised manuscript.
>
> > 2. Also, in the empirical study, the algorithm presented in the paper has not been compared to [A].
>
> **Reply:** Thank you for your insightful comment. Our empirical studies primarily focus on comparing the runtime of various algorithms in terms of their convergence to the global minimizer of a common convex log-determinant program, as illustrated in Figures 2-5 of our paper. **We clarify that a direct comparison with paper [A] is not feasible**, as its algorithm reaches a stationary point of its nonconvex formulation problem, which is fundamentally different from our approach.
>
> However, **we have made an indirect comparison with paper [A] in our first submission through our comparison with the PGD method**, which paper [A] employs to solve a log-determinant program at each stage. Our extensive experiments demonstrate that our algorithm significantly improves computational efficiency compared to PGD. Moreover, since paper [A]'s method requires the repeated use of PGD across different stages, presents a considerable advantage in terms of computational efficiency.
>
> To further address your concern, **we provide below an additional comparison with paper [A] in the experiment on financial time-series data** (which is presented in Appendix A). We compared both the graph edge recovery performance and runtime, using the modularity metric for edge recovery, where a higher value indicates a more accurate graph representation.
>
> The table below provides the modularity values and runtime for the method in paper [A], FPN, and FPN($\mathcal{E}$), where the latter two are proposed in our paper. The method in paper [A] leads to a higher modularity value than FPN due to its iterative refinement strategy. FPN($\mathcal{E}$), which imposes a disconnected set, yields a modularity value comparable to that of the method in [A]. In terms of computational time, both FPN and FPN($\mathcal{E}$) significantly outperform the method in paper [A], due to the relative inefficiency of the PGD used in [A] and the repeated need to solve the log-determinant program.
>
> | Methods |  paper [A]   |  FPN  |  FPN($\mathcal{E}$)  |
> |-------|------|------|------|
> | **Modularity** | 0.667 | 0.646 | 0.672 |
> | **Runtime** | 215.57 | 6.80 |  10.17 |
>
> **The graphs learned via the three methods are displayed in the attached PDF.**
>
>  We will include this detailed explanation and additional experimental results in our revised manuscript. We appreciate your feedback, which will undoubtedly enhance the quality of our paper.

---

> > ### Comment · Reviewer_7MEo · 2023-08-17
> > **Thank you for your response**
> >
> > Thank you for your response. I will consider your feedback.

---

> > > ### Author Response · Authors · 2023-08-18
> > >
> > > We sincerely appreciate the time and effort you've dedicated to reviewing our paper and considering our responses. Please be assured that we have concentrated our efforts to thoroughly address your concerns about paper [A], through both the individual response above and the global response.
> > >
> > > If you find our responses to be satisfactory, it would be our honor if you could consider increasing the review score. We warmly welcome any further questions or concerns that you may have.

---

### Official Review · Reviewer_AisR · 2023-07-07

**Soundness:** 4 excellent
**Presentation:** 4 excellent
**Contribution:** 3 good
**Rating:** 7
**Confidence:** 4

**Summary:**

The authors propose and analyze a variable metric projected gradient method for a variant of the graphical Lasso problem where the off-diagonal elements are additionally constraint to be non-positive, and in which specific entries of the precision matrix are known to be zero.
The proposed metric which is used to precondition the gradients comes from the curvature of the function $X \mapsto -\operatorname{log det}(X)$ , which is part of the objective, and thus incorporates second-order information of the objective function, unlike more standard approaches for the problem.

The authors put their result in the context of the recent literature, and show in numerical experiments on covariances created from multivariate Gaussian whose precision matrix sparsity comes from a Barabási-Albert random graph that the proposed method outperforms other methods used to optimize the same objective function, such as block coordinate descent or the proximal point algorithm in terms of clock-time convergence speed.

From a theoretical point of view, the authors show that the proposed algorithm indeed converges to the unique minimizer of the (convex) objective, that the proposed stepsize rule indeed works, and that the algorithm identifies the true support in a finite number of iterations under an additional reasonable assumption on the minimizer.

**Strengths:**

I consider the proposed efficient estimator for the sparse precision matrix with off-diagonal non-negativity constraint of interest for the machine learning community due to wide potential applicability of the setting to engineering problems.
The specific gradient preconditioning through the weighted metric induced by the second derivatives of the $\ell_2$-error is original for the problem, and the computational advantages compared to state-of-the-art methods are well explained ($O(p^3)$ time complexity) and showcased in sufficiently timing experiments. The choice of the comparison methods is adequate and the experimental setups are informative.

The paper is clearly and well written. A strength of the paper is the suitable co-design that heavily takes into account the problem structure, which makes the Hessian approximation used in the variable metric preconditioning simple enough to compute in a reasonable amount of time, while at the same time delivering a benefit in per-iteration improvement.

**Weaknesses:**

While the problem as formulated in (1) is convex as stated, the underlying estimator pipeline is somewhat inherently non-convex due to the fact that the $\lambda_{ij}$ correspond to weights of a reweighted $\ell_1$-term whose weights are chosen based on the MLE. This detail could be mentioned more prominently in the paper.

From a computational point of view, the line search method deployed has the computational disadvantage that a lot of function evaluations, which might be expensive due to their need for spectral decompositions, might be needed to make the method work.

It remains unclear whether the convergence rate result Theorem 4.5 is optimal. Due to the incorporation of second-order information, a discussion of why superlinear convergence cannot be shown or observed would be further informative.

**Questions:**

- When discussing the positive definiteness constraint in line 112, you mention that it is not closed and furthermore, cannot be managed by a projection. However, by looking at (1), it seems that due to the objective, it is clear that the minimizer will not be singular due to the existence of the term $- \operatorname{log det}(X)$  in the minimization: If at least one of the eigenvalues of X is 0, $\operatorname{log det}(X)$ is finite. Lemma B.1 touches upon this, but couldn't the positive definiteness constraint be relaxed to a psd constraint?
- In the considered model (1), the role of the disconnectivity constraint remains slightly unclear. Appendix A.1. and also the discussion of the financial time series example leaves the impression that the disconnectivity set is found by a similar method as the weights of the $\ell_1$-term $\lambda_{ij}$. Can you make the distinction clearer?
 - In Theorem 4.2, it will be convenient if one the authors can refer definition equation for $L_f$.

**Limitations:**

The this discussion of limitations is adequate in general. However, it would have been beneficial if the "hidden - $\lambda$-dependence" on the sparsity of the estimated covariance matrix had been explained better in the paper.

---

> ### Author Rebuttal · Authors · 2023-08-08
>
> > 1. From a computational point of view, the line search method deployed has the computational disadvantage that a lot of function evaluations, which might be expensive due to their need for spectral decompositions, might be needed to make the method work.
>
> **Reply:** Thank you for your perceptive observations. We acknowledge the potential computational intensity of the line search procedure, due to the need to compute the log-determinant function and verify positive definiteness. We genuinely appreciate your attention to this critical aspect.
>
> In our current implementation, we first conduct the Cholesky decomposition $\mathit{X} = \mathit{L} \mathit{L}^\top$ using MATLAB's "chol" function. This function can simultaneously verify the positive definiteness of $\mathit{X}$. Next, we calculate the log-determinant function as $\log \det (\mathit{X}) = 2 \sum_i \log ( L_{ii} )$. The Cholesky decomposition is the most computationally demanding step, generally requiring $O(p^3)$ costs for a $p \times p$ matrix. However, the sparsity of the matrix can substantially reduce the computational cost in MATLAB, as the "chol" function utilizes CHOLMOD. This function can process a sparse positive definite matrix of 10,000 dimensions within approximately 2 seconds. Nevertheless, the computational load could rise significantly as we scale to higher dimensions.
>
> To address the computational challenge, we draw attention to a more efficient method for evaluating the log-determinant function and verifying positive definiteness, as presented in [R1]. This method, leveraging Schur complements and sparse linear system solving, can tackle problems of up to 1,000,000 dimensions. Furthermore, it's worthwhile to investigate more efficient strategies for computing an approximate log-determinant function. In this context, the approach proposed in [R2] ensures a nearly linear scaling of execution time with the number of non-zero entries, while maintaining a high level of accuracy.
>
> This discussion will be integrated into our revised manuscript.
>
> References:
>
> [R1] C.-J. Hsieh, M. A. Sustik, I. S. Dhillon, P. K. Ravikumar, and R. Poldrack. "BIG & QUIC: Sparse inverse covariance estimation for a million variables." Advances in neural information processing systems, 2013.
>
> [R2] I. Han, D. Malioutov, and J. Shin. "Large-scale log-determinant computation through stochastic Chebyshev expansions." International Conference on Machine Learning, 37:908-917, 2015.
>
> > 2. It remains unclear whether the convergence rate result Theorem 4.5 is optimal. A discussion of why superlinear convergence cannot be shown or observed would be further informative.
>
> **Reply:** We appreciate your insightful comment. We would like to clarify that our convergence rate result is not optimal, as our algorithm generally does not attain superlinear convergence, despite the incorporation of second-order information. Superlinear convergence requires the inverse gradient scaling matrix to progressively approximate the Hessian at the minimizer, a condition that our algorithm does not satisfy.
>
> However, constructing a search direction to achieve superlinear convergence is significantly more computationally demanding than our method, as it cannot leverage the special structure of the Hessian to decrease the computational burden.
>
> We will include this discussion in the revised manuscript for further clarity.
>
> > 3. Couldn't the positive definiteness constraint be relaxed to a PSD constraint?
>
> **Reply:** Thank you for your insightful comment. We acknowledge your suggestion that the positive definiteness constraint can be relaxed to a positive semi-definiteness (PSD) constraint. This would indeed create a closed set that could be managed through projection.
>
> However, we chose not to follow this route due to the computational burden associated with projecting onto the intersection of the two sets - corresponding to the PSD and non-positive constraints. Implementing such a procedure, for example via Dykstra's projection algorithm, would involve alternating projection onto each set until convergence is achieved. This would significantly increase the computational load compared to our current method, which ensures positive definiteness through a line search method and manages the non-positive constraint via projection.
>
> We will enhance the revised manuscript with a more detailed explanation of this aspect.
>
> > 4. In the considered model (1), the role of the disconnectivity constraint remains slightly unclear. Can you make the distinction clearer?
>
> **Reply:** Thank you for your valuable feedback. Both the disconnectivity set and the $\ell_1$-term weights are derived based on the Maximum Likelihood Estimation (MLE) in our experiments, but they are procured through distinct techniques and serve different roles. The disconnectivity set is identified by applying hard thresholding to the MLE, and it is used to incorporate prior insights about the graph structure. On the other hand, the $\ell_1$-term weights, which are derived from a monotonically decreasing function of the MLE, are designed to promote sparsity in the solution.
>
> It's worth noting that the disconnectivity set and the $\ell_1$-term weights can be determined through various methodologies. In addition to the MLE, other estimators can also be effectively employed. For instance, one could use the thresholded sample covariance matrix supported by various thresholding techniques. Moreover, the disconnectivity set can also be obtained in the context of learning structured graphs, such as bipartite graphs, by incorporating inherent disconnection in graphs.
>
> We will provide a more detailed distinction between these two aspects in the revised manuscript.
>
> > 5. In Theorem 4.2, it will be convenient if the authors can refer definition equation for $L_f$.
>
> **Reply:** Thank you for your helpful suggestion. We will refer definition equation for $L_f$ in the revised manuscript to make it more convenient for readers.

---

> > ### Comment · Reviewer_AisR · 2023-08-16
> >
> > I thank the authors for their response, which clarifies in particular the computational aspects involving the line search and Cholesky decomposition. We recommend that the indicated remarks will indeed be included into the final manuscript.
> >
> > Following up on the discussion of the PSD constraint: It seems then to me that working with the PSD set instead of the PD set would not change anything in the algorithm if we kept the line search, as the positive definiteness is automatically enforced due to the form of the objective $f(\mathbf{X})$. If that is the case, we recommend mentioning that.
> >
> > Regarding the point about superlinear convergence: Could you elaborate on what the algorithm would look like if the exact Hessian was taken, and quantify how this makes the computational burden considerably worse? Maybe I am missing something, but the formula of Proposition 3.1. seems to be computationally of the same order as the formula of Proposition 3.3. Thank you!

---

> > > ### Author Response · Authors · 2023-08-16
> > >
> > > We sincerely appreciate your thorough review and insightful comments.
> > >
> > > You are correct that using the PSD set, rather than the PD set, would not alter the algorithm if we continue to use the line search method to handle it. We recognize the importance of this clarification and will ensure to include this point in the final manuscript.
> > >
> > > In response to your query regarding the use of the exact Hessian, if we were to employ the exact Hessian, specifically the Newton direction from Proposition 3.1, the algorithm's convergence to the minimizer cannot be guaranteed, despite its computational efficiency. This is primarily due to the fact that such a direction may not always serve as a descent direction. As depicted in Figure 1(c) of our paper, the Newton direction from Proposition 3.1 stops decreasing the objective function value after a few iterations, thereby failing to converge to the minimizer.
> > >
> > > If our goal is superlinear convergence, we would need to replace $[H _k^{-1}] _{\mathcal{I} _k^c\mathcal{I} _k^c}$ with $\left([H _k] _{\mathcal{I} _k^c\mathcal{I} _k^c}\right)^{-1}$ in equation (11). However, it is important to note that this modification would significantly increase the computational burden compared to our method, as it cannot leverage the special structure of the Hessian to lessen the computational load. Moreover, the size of $\mathcal{I}_k^c$ could become large during iterations, which is not within our control. This could lead to a per-iteration complexity of up to $O(p^6)$. In contrast, our method guarantees a complexity of $O(p^3)$ per iteration.
> > >
> > > We hope these explanations sufficiently address your inquiries. We once again express our sincere gratitude for your valuable feedback. To further enhance the clarity of our work, we plan to incorporate all these discussions related to line search, Cholesky decomposition, the exact Hessian, and superlinear convergence into the revised manuscript.

---

### Author Rebuttal · Authors · 2023-08-09

Dear Reviewers,

We sincerely thank you for dedicating your time to review our paper and for your insightful comments. These have significantly contributed to improving the clarity and overall quality of our manuscript.

In this global response, we embrace the opportunity to clarify the distinction between our paper and paper [A], and our unique contributions relative to [A]. This issue was raised by one of the reviewers. It is worth mentioning that Paper [A] was published after an earlier version of our work had already been posted on arXiv and it cites this earlier version.

While both papers aim to advance the field of MTP2 graphical models, they do so with different emphases and distinct methodologies, thus offering different contributions.

### 1. Divergent Emphases and Methodologies Between Our Paper and Paper [A]:

Paper [A] primarily targets the design of an estimator that can concurrently achieve superior performance in precision matrix estimation and graph edge recovery. **The authors' emphasis in paper [A] leans more towards enhancing estimation performance, even at the expense of computational efficiency.** They present a multiple-stage estimation method, founded on the framework of multiple convex relaxation, which progressively refines estimates across stages. This requires the repeated solving of the constrained log-determinant program across stages. Their theoretical analysis offers an upper bound of the estimation error between their proposed estimator and the underlying precision matrix, but does not establish a guarantee for the algorithm's convergence or the convergence rate. Their experimental results primarily demonstrate the performance of estimating precision matrices and recovering graph edges using metrics like relative estimation error, true positive rate, false positive rate, and F-score. However, this method is computationally challenging in high-dimensional scenarios due to the relatively slow convergence of the PGD and its repeated need to use PGD to solve the log-determinant program across stages.

In contrast, **our paper emphasizes the enhancement of computational efficiency, recognizing it as a critical requirement when dealing with large-scale problems.** Our focus is on developing an efficient and scalable algorithm tailored for estimating large-scale MTP2 graphical models. Both per-iteration complexity and convergence rate are of paramount importance in this endeavor. We propose a novel second-order algorithm, grounded in the two-metric projection method. This algorithm substantially reduces per-iteration complexity by leaving a special structure of the Hessian. It also achieves a rapid convergence rate by incorporating second-order information into the search direction. Our empirical study primarily compares the runtime of various algorithms in terms of their convergence to the global minimizer of a common convex log-determinant program, as depicted in Figures 2-5. It's important to note that all algorithms share the same estimation performance in these experiments, given that they all solve the same problem.

### 2. Unique Contributions of Our Paper:

Existing algorithms designed to solve the sign-constrained log-determinant program, such as Projected Gradient Descent (PGD) and Block Coordinate Descent (BCD), encounter computational challenges in high-dimensional scenarios due to their slow convergence rate or high per-iteration computational complexity. To mitigate these issues, **we propose a novel projected Newton-like algorithm, which achieves a significantly faster convergence rate than first-order algorithms while maintaining the same per-iteration complexity.** We provide solid evidence of its guaranteed convergence to the global minimizer and ascertain its theoretical convergence rate. Our extensive experiments demonstrate that our proposed algorithm significantly enhances computational efficiency compared to state-of-the-art methods.

It is worth noting that paper [A] employs the PGD algorithm to tackle the constrained log-determinant program at each stage. However, being a first-order method, it is often hindered by computational inefficiency due to its slow convergence rate. In contrast, **our proposed algorithm can boost the computational efficiency of the method outlined in paper [A]** by applying it to solve the log-determinant program across stages, thereby enabling its application to higher-dimensional problems. However, it's important to acknowledge that PGD, as demonstrated in paper [A], has a distinct advantage in handling complex constraints, such as the diagonally dominant M-matrices, a capability currently not supported by our algorithm.

In the revised manuscript, we will ensure to appropriately cite paper [A] and clearly articulate its relevance to our work. Once again, we express our gratitude to all reviewers for your valuable feedback, which greatly contributes to improving the clarity and thoroughness of our manuscript.

Reference:

[A] J. Ying, J. V. D. M. Cardoso, and D. P. Palomar. "Adaptive Estimation of Graphical Models under Total Positivity." ICML, 2023.

---

### Decision · Program_Chairs · 2023-09-21

**Decision:**

Accept (poster)

**Comment:**

In this paper, the authors considered the problem of sparse precision matrix estimation under total positivity and developed a fast projected Newton-type algorithm for efficiently solving the formulated sign-constrained log-determinant program. After discussions, the reviewers reached a majority consensus that the proposed algorithm is theoretically sound and numerically convincing. One of the reviewers expressed a major concern regarding the missing comparison to a closely relevant work, which has been thoroughly addressed by the authors in their responses. Based on the reviews, authors responses and post-rebuttal discussions, it was assessed that this submission makes a sufficiently novel and solid contribution to the addressed problem and thus is worthy of publication.